# The SARS-CoV-2 nucleocapsid protein is dynamic, disordered, and phase separates with RNA

Jasmine Cubuk [1,2], Jhullian J. Alston [1,2], J. Jeremías Incicco[1,2], Sukrit Singh [1,2],
Melissa D. Stuchell-Brereton[1,2], Michael D. Ward[1,2], Maxwell I. Zimmerman[1,2], Neha Vithani[1,2],
Daniel Griffith [1,2], Jason A. Wagoner[3], Gregory R. Bowman[1,2], Kathleen B. Hall[1], Andrea Soranno [1,2✉] &
Alex S. Holehouse [1,2✉]

The SARS-CoV-2 nucleocapsid (N) protein is an abundant RNA-binding protein critical for viral genome packaging, yet the molecular details that underlie this process are poorly understood. Here we combine single-molecule spectroscopy with all-atom simulations to uncover the molecular details that contribute to N protein function. N protein contains three dynamic disordered regions that house putative transiently-helical binding motifs. The two folded domains interact minimally such that full-length N protein is a flexible and multivalent RNA-binding protein. N protein also undergoes liquid-liquid phase separation when mixed with RNA, and polymer theory predicts that the same multivalent interactions that drive phase separation also engender RNA compaction. We offer a simple symmetry-breaking model that provides a plausible route through which single-genome condensation pre-ferentially occurs over phase separation, suggesting that phase separation offers a convenient macroscopic readout of a key nanoscopic interaction.

[1] Department of Biochemistry and Molecular Biophysics, Washington University School of Medicine, St. Louis, MO, USA. [2] Center for Science and Engineering of Living Systems (CSELS), Washington University in St. Louis, St. Louis, MO, USA. [3] Laufer Center for Physical and Quantitative Biology, Stony Brook University, Stony Brook, NY, USA. ✉email: soranno@wustl.edu; alex.holehouse@wustl.edu

Severe acute respiratory syndrome coronavirus 2 (SARS-CoV-2) is an enveloped, positive-strand RNA virus that causes the disease COVID-19 (Coronavirus Disease-2019)[1]. While coronaviruses typically cause relatively mild respiratory diseases, as of February 2021 COVID-19 is on course to kill 2.5 million people since its emergence in late 2019 [1–3]. While recent progress in vaccine development has been remarkable, the emergence of novel coronaviruses in human populations represents a continuing threat[4]. As a result, therapeutic approaches that address fundamental and general viral mechanisms will offer a key route for first-line intervention against future pandemics.

A challenge in identifying candidate drugs is our relatively sparse understanding of the molecular details that underlie the function of SARS-CoV-2 proteins. As a result, there is a surge of biochemical and biophysical exploration of these proteins, with the ultimate goal of identifying proteins that are suitable targets for disruption, ideally with insight into the molecular details of how disruption could be achieved[5,6].

While much attention has been focused on the Spike (S) protein, many other SARS-CoV-2 proteins play equally critical roles in viral physiology, yet we know relatively little about their structural or biophysical properties[7–10]. Here we performed a high-resolution structural and biophysical characterization of the SARS-CoV-2 nucleocapsid (N) protein, the protein responsible for genome packaging[11,12]. A large fraction of N protein is predicted to be intrinsically disordered, which constitutes a major barrier to conventional structural characterization[13]. To overcome these limitations, we combined single-molecule spectroscopy with all-atom simulations to build a residue-by-residue description of all three disordered regions in the context of their folded domains. The combination of single-molecule spectroscopy and simulations to reconstruct structural ensembles has been applied extensively to uncover key molecular details underlying disordered protein regions[14–19]. Our goal here is to provide biophysical and structural insights into the physical basis of N protein function.

In exploring the molecular properties of N protein, we discovered that it undergoes phase separation with RNA, as was also reported recently[20–27]. Given N protein underlies viral packaging, we reasoned phase separation may in fact be an unavoidable epiphenomenon that reflects physical properties necessary to drive the compaction of long genomic RNA molecules. To explore this principle further, we developed a simple physical model, which suggested symmetry breaking through a small number of high-affinity-binding sites that can organize anisotropic multivalent interactions to drive single-polymer compaction, as opposed to multi-polymer phase separation. Irrespective of its physiological role, our results suggest that phase separation provides a macroscopic readout (visible droplets) of a nanoscopic process (protein:RNA and protein:protein interaction). In the context of SARS-CoV-2, those interactions are expected to be key for viral packaging, such that assays that monitor phase separation of N protein with RNA may offer a convenient route to identify compounds that will also attenuate viral assembly.

## Results

Coronavirus nucleocapsid proteins are multi-domain RNA-binding proteins that play a critical role in many aspects of the viral life cycle[12,28]. The SARS-CoV-2 N protein shares substantial sequence conservation with other coronavirus nucleocapsid proteins (Figs. S1–5). Work on N protein from a range of model coronaviruses has shown that N protein undergoes both self-association, interaction with other proteins, and interaction with RNA, all in a highly multivalent manner.

The SARS-CoV-2 N protein can be divided into five domains: a predicted intrinsically disordered N-terminal domain (NTD), an RNA-binding domain (RBD), a predicted disordered central linker (LINK), a dimerization domain, and a predicted disordered C-terminal domain (CTD) (Fig. 1A–C). While SARS-CoV-2 is a novel coronavirus, decades of work on model coronaviruses (including SARS coronavirus) have revealed a number of features expected to hold true in the SARS-CoV-2 N protein. Notably, all five domains are predicted to bind RNA[29–35], and while the dimerization domain facilitates the formation of well-defined stoichiometric dimers, RNA-independent higher-order oligomerization is also expected to occur[34,36–38]. Importantly, protein–protein and protein–RNA interaction sites have been mapped to all three disordered regions.

Despite recent structures of the RBD (Fig. 1B) and dimerization domains (Fig. 1C) from SARS-CoV-2, the solution-state conformational behavior of the full-length protein remains elusive[39–41]. Understanding N protein function necessitates a mechanistic understanding of the flexible predicted disordered regions and their interplay with the folded domains. A recent small-angle X-ray study shows good agreement with previous work on SARS, suggesting the LINK is relatively extended, but

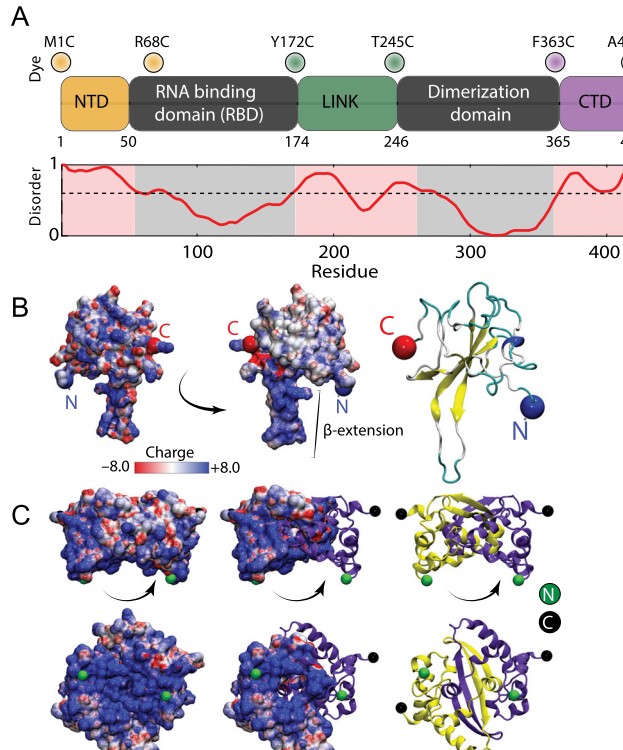

**Fig. 1 Sequence and structural summary of N protein. A** Domain architecture of the SARS-CoV-2 N protein with disorder prediction performed using IUPred2A[142]. Dye positions used in this study are annotated across the top, disorder prediction calculated across the bottom. The specific positions were selected such that fluorophores are sufficiently close to be in the dynamic range of FRET measurements. Labeling was achieved using cysteine mutations and thiol-maleimide chemistry. **B** Structure of the SARS-CoV-2 RNA-binding domain (RBD) (PDB: 6yi3). Center and left: colored based on surface potential calculated with the Adaptive Poisson Boltzmann Method[143], revealing the highly basic surface of the RBD. Right: ribbon structure with N- and C-termini highlighted. **C** Dimer structure of the SARS-CoV-2 dimerization domain (PDB: 6yun). Center and left: colored based on surface potential, revealing the highly basic surface. Right: ribbon structure with N- and C-termini highlighted.

neither the structural basis for this extension nor the underlying dynamics are known[29,42].

Here, we address these questions by probing three full-length constructs of the N protein with fluorescent labels (Alexa 488 and 594) flanking the NTD, the LINK, and the CTD (see Fig. 1A and Table S1). These constructs allow us to quantify conformations and dynamics of the disordered regions in the context of the full-length protein using single-molecule Förster resonance energy transfer (FRET) and fluorescence correlation spectroscopy (FCS) (see SI for details). We also investigated the stability of the RBD and truncated variants of the protein to test the role of long range interactions on the disordered regions (see SI and Table S2). In parallel to the experiments, we performed all-atom Monte Carlo simulations of each of the three IDRs in isolation and in context with their adjacent folded domains.

**The NTD is disordered, flexible, and transiently interacts with the RBD.** We started our analysis by investigating the NTD conformations. Under native conditions, single-molecule FRET measurements revealed the occurrence of a single population with a mean transfer efficiency of $0.65 \pm 0.03$ (Figs. 2A and S6). To assess whether this transfer efficiency reports on a rigid distance (e.g., structure formation or persistent interaction with the RBD) or is a dynamic average across multiple conformations, we first compare the lifetime of the fluorophores with transfer efficiency. Under native conditions, the donor and acceptor lifetimes for the NTD construct lie on the line that represents fast conformational dynamics (Fig. S7A). To properly quantify the timescale associated with these fast structural rearrangements, we leveraged nanosecond FCS. As expected for a dynamic population[43,44], the cross-correlation of acceptor–donor photons for the NTD is anticorrelated (Figs. 2B and S12). A global fit of the donor–donor, acceptor–acceptor, and acceptor–donor correlations yields a reconfiguration time $\tau_r = 170 \pm 30$ ns. This is longer than reconfiguration times observed for other proteins with a similar persistence length and charge content[44–47], hinting at a large contribution from internal friction due to rapid intramolecular contacts (formed either within the NTD or with the RBD) or transient formation of short structural motifs[48]. A conversion from transfer efficiency to chain dimensions can be obtained by assuming the distribution of distances computed from polymer models. Assuming a Gaussian chain distribution yields a root-mean-square distance between the fluorophores $r_{1-68}$ of $48 \pm 2$ Å. When using the recently proposed self-avoiding walk (SAW) model[49] (see Supplementary Information), we compute a value of $r_{1-68}$ $47 \pm 2$ Å. This corresponds to values of persistence length (see SI) equal to $4.5 \pm 0.4$ and $4.3 \pm 0.4$ Å for the Gaussian and SAW distribution, respectively, which are similar to values reported for another unfolded protein under native conditions[44–46,50]. Overall, these results confirm the NTD is disordered, as predicted by sequence analysis.

We next examined the interaction of the NTD with other domains in the protein. We studied a truncated N protein variant that contains only the NTD and RBD domains (NTD–RBD) and samples identical labeling positions. The root-mean-square distance $r_{1-68}$ is $46 \pm 2$ Å for both the Gaussian and SAW models, within errors from the NTD-FL values, suggesting no or limited interaction between the NTD and the LINKER, DIMER, and CTD domains (see Fig. S8 and Table S2). We then assessed the role of the folded RBD and its influence on the conformations of the NTD by studying the effect of a chemical denaturant on the protein. The titration with guanidinium chloride (GdmCl) reveals a decrease of transfer efficiencies when moving from native buffer conditions to 1 M GdmCl, followed by a plateau of the transfer efficiencies at concentrations between 1 and 2 M and a

subsequent further decrease at higher concentrations (Figs. S6 and S8). This behavior can be understood assuming that the plateau between 1 and 2 M GdmCl represents the average of transfer efficiencies between two populations in equilibrium that have very close transfer efficiency and are not completely resolved because of shot noise. We interpret these two populations as the contribution of the folding and unfolding fraction of the RBD domain on the distances probed by the NTD-FL construct, which includes a labeling position within the folded RBD. Indeed, this interpretation is supported by a broadening in the transfer efficiency peak between 1 and 2 M GdmCl. Besides the effect of the unfolding of the RBD, the dimensions of the NTD-FL are also modulated by a change in the solvent quality when adding denaturant (Figs. 2C, S6 and S8) and this contribution to the expansion of the chain can be described using an empirical binding model[51–55]. A fit of the interdye root-mean-square distances to this model and the inferred stability of the RBD domain (midpoint: $1.3 \pm 0.2$ M; $\Delta G_0 = (5 \pm 1)$ kcal mol$^{-1}$) are presented in Fig. 2C. A comparative fit of the histograms assuming two overlapping populations yields a consistent result in terms of RBD stability and protein conformations (Fig. S9). To confirm the inferred RBD stability results, we directly interrogated the RBD domain by measuring a full-length construct with labels in position 68 and 172, which flanks the folded RBD structure (see section "RBD folding" in SI). Though the denaturation of the RBD reveals coexistence of up to three populations, which we identify as an unfolded, an intermediate, and a folded state, the range of the folding transition is compatible with the estimates made using the NTD constructs (midpoint: $1.68 \pm 0.02$ M, see Fig. S9 and Table S6).

To better understand the sequence-dependent conformational behavior of the NTD, we turned to all-atom simulations of an NTD–RBD construct. We used a novel sequential sampling approach that integrates long timescale MD simulations performed using the Folding@home distributed computing platform with all-atom Monte Carlo simulation performed with the ABSINTH forcefield to generate an ensemble of almost 400,000 distinct conformations (see "Methods")[56–58]. We also performed simulations of the NTD in isolation.

We observed good agreement between simulation and experiment for the equivalent inter-residue distance (Fig. 2D). The peaks on the left side of the histogram reflect specific simulations where the NTD engages more extensively with the RBD through a fuzzy interaction, leading to local kinetic traps[59]. We also identified several regions in the NTD where transient helices form, and using normalized distance maps found regions of transient attractive and repulsive interaction between the NTD and the RBD (Fig. 2E). In particular, the basic beta-strand extension from the RBD (Fig. 1B) repels the arginine-rich C-terminal region of the NTD, while a phenylalanine residue (F17) in the NTD engages with a hydrophobic face on the RBD (Fig. 2G). Finally, we noticed the arginine-rich C-terminal residues (residues 31–38) form a transient alpha helix projecting three of the four arginines in the same direction (Figs. 2F, H). These features provide molecular insight into previously reported functional observations (see "Discussion").

**The linker is highly dynamic and there is minimal interaction between the RBD and the dimerization domain.** We next turned to the linker (LINK FL) construct to investigate how the disordered region modulates the interaction and dynamics between the two folded domains. Under aqueous buffer conditions, single-molecule FRET reveals the coexistence of two overlapping populations with mean transfer efficiencies of $0.55 \pm 0.03$ and $0.75 \pm 0.03$, respectively (Fig. 3A). A small change in

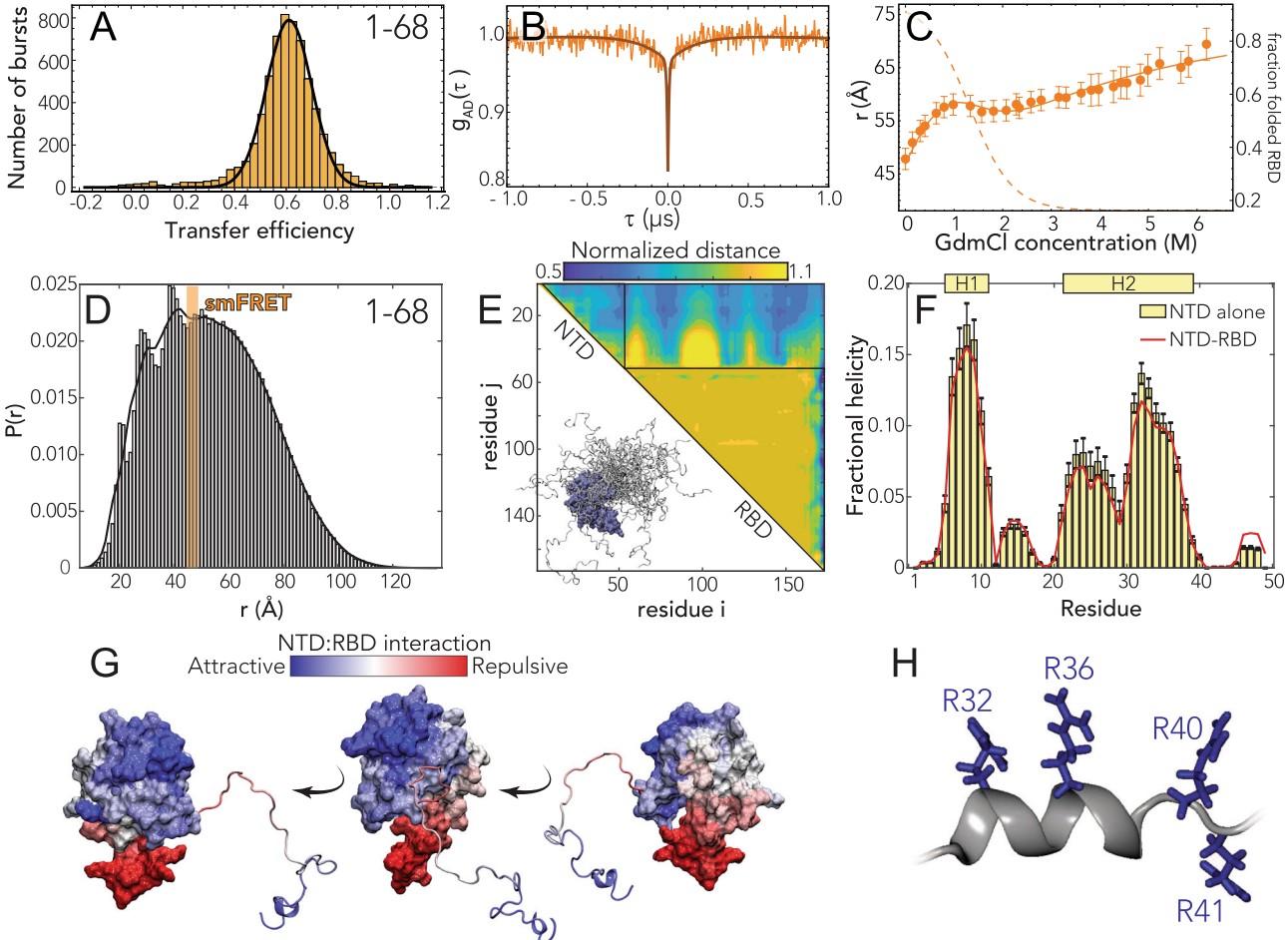

**Fig. 2 The N-terminal domain (NTD-FL) is disordered with residual helical motifs. A** Histogram of the transfer efficiency distribution measured across the labeling positions 1 and 68 in the context of the full-length protein, under aqueous buffer conditions (50 mM Tris buffer). **B** Donor–acceptor cross-correlation measured by ns-FCS (see SI). The observed anticorrelated rise is the characteristic signature of FRET dynamics and the timescale associated is directly related to the reconfiguration time of the probed segment. **C** Root-mean-square interdye distance as extracted from single-molecule FRET experiments across different denaturant concentrations using a Gaussian chain distribution, examining residues 1–68 in the context of the full-length protein. The full line represents a fit to the model in Eq. (S7), which accounts for denaturant binding (see Table S2) and unfolding of the folded RBD. The dashed line represents the estimated fraction of folded RBD across different denaturant concentrations based on Eq. (S8). Error bars represent propagation ±0.03 systematic error in measured transfer efficiencies (see SI). **D** All-atom simulations of the NTD in the context of RBD reveal good agreement with smFRET-derived average distances. The peaks on the left shoulder of the histogram are due to persistent NTD–RBD interactions in a small subset of simulations. **E** Normalized distance maps (scaling maps) quantify heterogeneous interaction between every pair of residues in terms of average inter-residue distance normalized by distance expected for the same system if the IDR had no attractive interactions (the excluded volume limit[144]). Both repulsive (yellow) and attractive (blue) regions are observed for NTD–RBD interactions. **F** Transient helicity (residues 5–11 and 21–39) in the NTD in isolation or in the context of the RBD. Perfect profile overlap suggests interaction between the NTD and the RBD does not lead to a loss of helicity. Error bars are standard error of the mean calculated from forty independent simulations. **G** Projection of normalized distances onto the folded domain reveals repulsion is through electrostatic interaction (positively charged NTD is repelled by the positive face of the RBD, which is proposed to engage in RNA binding) while attractive interactions are between positive, aromatic, and polar residues in the NTD and a slightly negative and hydrophobic surface on the RBD (see Fig. 1B, center). **H** The C-terminal half of the transient helix H2 encodes an arginine-rich surface.

ionic strength of the solution is sufficient to alter the equilibrium between these two populations and favor the low transfer efficiency state (see inset in Fig. 3C). Comparison of the fluorescence lifetimes and transfer efficiencies indicates that, like the NTD, the transfer efficiencies represent dynamic conformational ensembles sampled by the LINK (Fig. S7A). ns-FCS confirms fast dynamics across the measured distribution of transfer efficiencies, with a characteristic reconfiguration time $\tau_r$ of $120 \pm 20$ ns (Figs. 3B and S12). This reconfiguration time is compatible with high internal friction effects, as observed for other unstructured proteins[44,45], but may also account for the drag of the surrounding domains. The root-mean-square interdye distance corresponding to the low

transfer efficiency population $r_{172-245}$ is equal to $55 \pm 2$ Å ($l_p = 5.4 \pm 0.4$ Å) when assuming a Gaussian chain distribution and 54 $\pm 2$ Å ($l_p = 5.2 \pm 0.4$ Å) when using a SAW model (see SI). In contrast, the root-mean-square interdye distance corresponding to the high transfer efficiency population is equal to $42 \pm 2$ Å when assuming a Gaussian Chain distribution or $45 \pm 2$ Å using the SAW model (with a corresponding $l_p = 3.2 \pm 0.3$ Å and $l_p = 3.6 \pm 0.3$ Å, respectively) (see SI).

Next, we addressed whether the LINK segment populates elements of persistent secondary structure or forms stable interaction with the RBD or dimerization domains. The addition of denaturant leads to the rapid loss of the high transfer efficiency

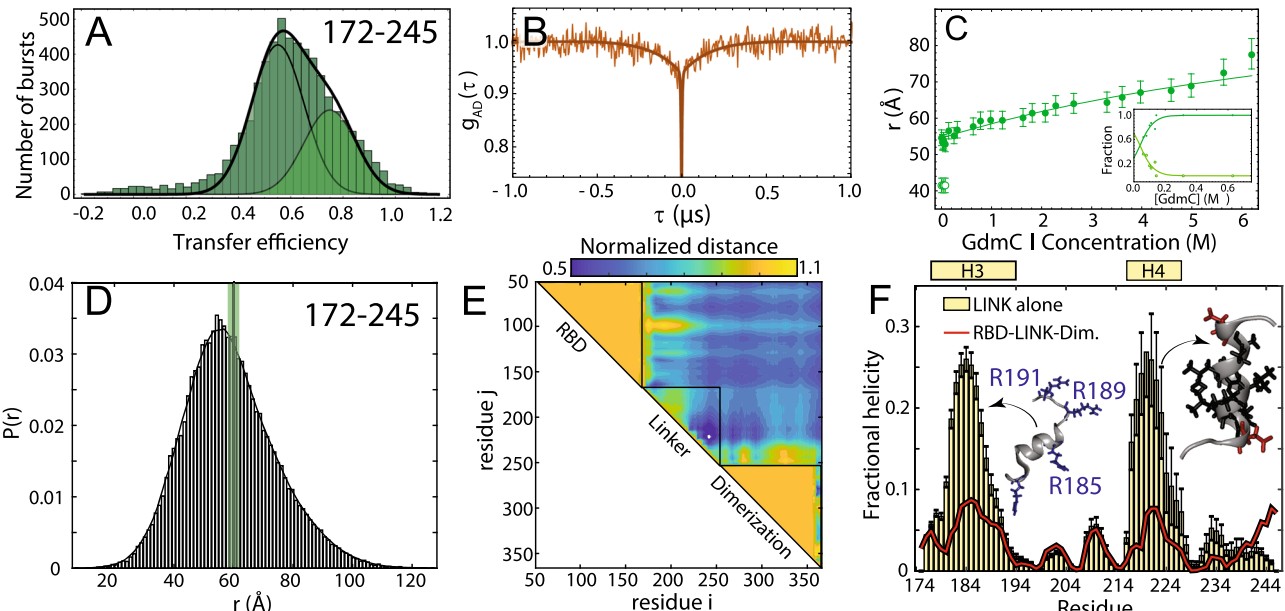

**Fig. 3 The RNA-binding domain (RBD) and dimerization domains are interconnected by a flexible disordered linker (LINK). A** Histogram of the transfer efficiency distribution measured across the labeling positions 172 and 245 in the context of the full-length protein, under aqueous buffer conditions. **B** Donor–acceptor cross-correlation measured by ns-FCS (see SI). The observed anticorrelated rise is the characteristic signature of FRET dynamics and the timescale associated is directly related to the reconfiguration time of the probed segment. **C** Interdye distance as extracted from single-molecule FRET experiments across different denaturant concentrations. The full line represents a fit to the model in Eq. (S6), which accounts for denaturant binding. The inset provides an estimate of the fraction of each population in the low GdmCl concentration regime. Error bars are the propagation of ±0.03 systematic error in measured transfer efficiencies (see SI). **D** Inter-residue distance distributions calculated from simulations (histogram) show good agreement with distances inferred from single-molecule FRET measurements (green bar). **E** Scaling maps reveal repulsive interactions between the N- and C-terminal regions of the LINK with the adjacent folded domains. We also observe relatively extensive intra-LINK interactions around helix H4 (see **F**). **F** Two transient helices are observed in the linker (residues 177–194 and 216–227). The N-terminal helix H3 overlaps with part of the SR region and orientates three arginine residues in the same direction, analogous to behavior observed for H2 in the NTD. The C-terminal helix H4 overlaps with a Leu/Ala-rich motif and may be a conserved nuclear export signal (see "Discussion"). Error bars are standard errors of the mean calculated from 30 independent simulations.

population and a continuous shift of the remaining population toward lower transfer efficiencies (Figs. S6 and S8). These results correspond to an almost linear expansion of the chain in response to denaturant (see Fig. 3C).

To better understand the nature of the two populations and explain the weak dependence of the linker expansion on denaturant, we investigated the same labeling positions in the absence of the DIMER and CTD domains (LINK ΔDIMER) (Table S2). smFRET measurements of this truncated version revealed a single population that undergoes a strong compaction with decreasing GdmCl concentration (Figs. S6 and S8). Interestingly the transfer efficiency measured in aqueous buffer is equivalent to the one reported by the high transfer efficiency population of the LINK FL construct. The electrostatic nature of this compaction is clearly revealed by titrating a polar non-ionic denaturant (urea) and observing that the chain remains largely compact and recovers the same dimensions measured in GdmCl only when adding salt to the solution (Fig. S10). Overall, the LINK ΔDIMER observations lead us to speculate that the LINK domain can either self-interact or interact with the RBD domain, whereas addition of the DIMER and CTD domains restricts these configurations and largely favor more expanded states with the exceptions of very low ionic strength conditions. To further explore the configurations of the LINK, we turned again to Monte Carlo simulations.

As with the NTD, all-atom Monte Carlo simulations provide atomistic insight that can be compared with our spectroscopic results. Given the size of the system, an alternative sampling strategy to the NTD–RBD construct was pursued here that did not include MD simulations of the folded domains, but we

instead performed simulations of a construct that included the RBD, LINK, and dimerization domain (but not the NTD and CTD). In addition, we also performed simulations of the LINK in isolation.

We again found good agreement between simulations and experiment (Fig. 3D). The root-mean-square inter-residue distance for the low transfer efficiency population (between simulated positions 172 and 245) is 59.1 Å, which is within the experimental error of the single-molecule observations. Normalized distance map shows a number of regions of repulsion, notably that the RBD repels the N-terminal part of the LINK and the dimerization domain repels the C-terminal part of the LINK (Fig. 3E). We tentatively suggest this may reflect sequence properties chosen to prevent aberrant interactions between the LINK and the two folded domains. In the LINK-only simulations we identified two regions that form transient helices at low populations (20–25%), although these are less prominent in the context of the full-length protein (Fig. 3F). These two helices encompass a serine–arginine (SR) rich region known to mediate both protein–protein and protein–RNA interaction. Helix H3 formation leads to the alignment of three arginine residues along one face of the helix. The second helix (H4) is a leucine/alanine-rich hydrophobic helix which may contribute to oligomerization, or act as a helical recognition motif for other protein interactions (notably as a nuclear export signal (NES) for Crm1, see "Discussion").

**The CTD engages in transient but non-negligible interactions with the dimerization domain.** Finally, we again applied

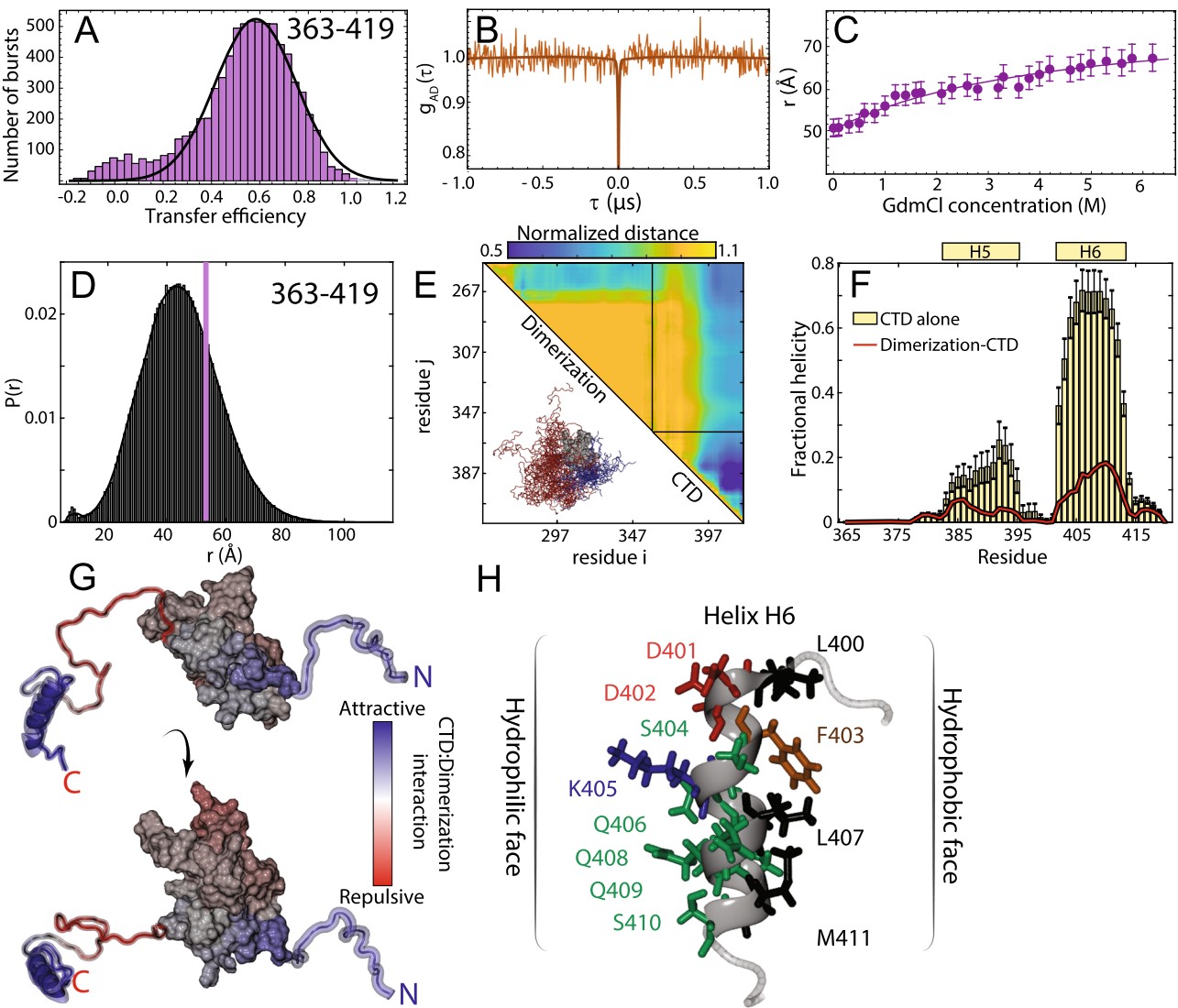

**Fig. 4 The C-terminal domain (CTD) is disordered, engages in transient interaction with the dimerization domain, and contains a putative helical-binding motif. A** Histogram of the transfer efficiency distribution measured across the labeling positions 363 and 419 in the context of the full-length protein, under aqueous buffer conditions. **B** Donor–acceptor cross-correlation measured by ns-FCS (see SI). The flat correlation indicates a lack of dynamics in the studied timescale or the coexistence of two populations in equilibrium whose correlations (one correlated and the other anticorrelated) compensate each other. **C** Interdye distance as extracted from single-molecule FRET experiments across different denaturant concentrations. The full line represents a fit to the model in Eq. (S6), which accounts for denaturant binding. Error bars are the propagation of ±0.03 systematic error in measured transfer efficiencies (see SI). **D** Inter-residue distance distributions calculated from simulations (histogram) show good agreement with distances inferred from single-molecule FRET measurements (purple bar). **E** Scaling maps describe the average inter-residue distance between each pair of residues, normalized by the distance expected if the CTD behaved as a self-avoiding random coil. H6 engages in extensive intra-CTD interactions and also interacts with the dimerization domain. We observe repulsion between the dimerization domain and the N-terminal region of the CTD. **F** Two transient helices (H5 and H6) are observed in the CTD (residues 383–396 and 402–415). Both show a reduction in population in the presence of the dimerization domain at least in part because the same sets of residues engage in transient interactions with the dimerization domain. Error bars are standard error of the mean calculated from forty independent simulations. **G** The normalized distances are projected onto the surface to map CTD-dimerization interaction. The helical region drives intramolecular interaction, predominantly with the N-terminal side of the dimerization domain. **H** Helix H6 is an amphipathic helix with a polar/charged surface (left) and a hydrophobic surface (right).

single-molecule FRET (Fig. 4A) and ns-FCS (Fig. 4B) to understand the conformational behavior of the CTD FL construct. Single-molecule FRET experiments again reveal a single population with a mean transfer efficiency of 0.59 ± 0.03 (Fig. 4A) and the denaturant dependence follows the expected trend for a disordered region, with a shift of the transfer efficiency toward lower values (Figs. 4C, S6 and S8), from 0.59 to 0.35. Interestingly, when studying the denaturant dependence of the protein, we noticed that the width of the distribution increases while moving toward aqueous buffer conditions. This suggests that the protein may

form transient contacts or adopt local structure. Comparison with a truncated variant that contains only the CTD (Fig. S8) reveals a very similar distribution, with almost identical mean transfer efficiency but a narrower width (Fig. S6), suggesting that part of the broadening is due to interactions with the neighboring domains.

To further investigate putative interaction between the CTD and neighboring domains, we turned to the investigation of protein dynamics. Though the comparison of the fluorophore lifetimes against transfer efficiency (Fig. S7A) appears to support

a dynamic nature underlying the CTD FL population, ns-FCS reveals a flat acceptor–donor cross-correlation on the nanosecond timescale (Fig. 4B). However, inspection of the donor–donor and acceptor–acceptor autocorrelations reveal a correlated decay. This is different from that expected for a completely static system such as polyprolines[60], where the donor–donor and acceptor–acceptor autocorrelation are also flat. An increase in the autocorrelations can be observed for static quenching of the dyes with aromatic residues. Interestingly, donor dye quenching can also contribute to a positive amplitude in the donor–acceptor correlation[61,62]. Therefore, a plausible interpretation of the flat cross-correlation data is that we are observing two populations in equilibrium whose correlations (one anticorrelated, reflecting conformational dynamics, and one correlated, reflecting quenching due contact formation) compensate each other.

To further investigate the possibility of two coexisting populations, we performed ns-FCS at increasing GdmCl concentrations. These experiments revealed a progressive increase of the anticorrelated amplitude in the cross-correlation, consistent with an increase of the dynamic population. Moreover, we also observed a simultaneous decrease in the overall donor–donor autocorrelation amplitude, consistent with a decrease in the quenched population (Fig. S12). Taken together, these results support our hypothesis that there are at least two distinct species existing in equilibrium. By analyzing the dynamic species between 0.16 and 0.6 M GdmCl, we quantified an average reconfiguration time ($\tau_r$) of $64 \pm 7$ ns for the dynamic population in the CTD. Under the assumption that the mean transfer efficiency still originates (at least partially) from a dynamic distribution, the estimate of the inter-residue root-mean-square distance is $r_{363-419} = 51 \pm 2$ Å ($l_p = 6.1 \pm 0.5$ Å) for a Gaussian chain distribution and $r_{363-419} = 48 \pm 1$ Å ($l_p = 5.4 \pm 0.4$ Å) for the SAW model (see SI). However, some caution should be used when interpreting these numbers since we know there is some contribution from fluorophore static quenching, which may in turn contribute to an underestimate of the effective transfer efficiency[63].

We again obtained good agreement between all-atom Monte Carlo simulations and experiments (Fig. 4D). Scaling maps reveal extensive intramolecular interaction by the residues that make up H6, both in terms of local intra-IDR interactions and interaction with the dimerization domain (Fig. 4E). We identified two transient helices, one (H5) is minimally populated but the second (H6) is more highly populated in the IDR-only simulation and still present at ~20% in the folded state simulations (Fig. 4F). The difference reflects the fact that several of the helix-forming residues interact with the dimerization domain, leading to a competition between helix formation and intramolecular interaction. Mapping normalized distances onto the folded structure reveals that interactions occur primarily with the N-terminal portion of the dimerization domain (Fig. 4G). As with the LINK and the NTD, a positively charged set of residues immediately adjacent to the folded domain in the CTD drive repulsion between this region and the dimerization domain. H6 is the most robust helix observed across all three IDRs, and is a perfect amphipathic helix with a hydrophobic surface on one side and charged/polar residues on the other (Fig. 4H). The cluster of hydrophobic residues in H6 engage in intramolecular contacts and offer a likely physical explanation for the complex ns-FCS data in aqueous buffer.

**N protein undergoes phase separation with RNA.** Over the last decade, biomolecular condensates formed through phase separation have emerged as a new mode of cellular organization[64–67]. Many of the proteins that have been shown to drive phase separation in vitro are RNA-binding proteins with intrinsically disordered regions[64,68]. Moreover, multivalency is the key molecular feature that determines if a biomolecule can undergo higher-order assembly[69]. Having characterized N protein to reveal three IDRs with distinct binding sites for both protein–protein and protein–RNA interactions it became clear that N protein possesses all of the features consistent with a protein that may undergo phase separation. With these results in hand, we anticipated that N protein would undergo phase separation with RNA[70–72].

In line with this expectation, we observed robust droplet formation with homopolymeric RNA (Fig. 5A, B) under aqueous buffer conditions, both at 50 mM Tris and at a higher salt concentration of 50 mM NaCl. Turbidity assays at different concentrations of protein and poly(rU) (200–250 nucleotides) demonstrate the classical re-entrant phase behavior expected for a system undergoing heterotypic interaction (Fig. 5C, D). It is to be noted that turbidity experiments do not exhaustively cover all the conditions for phase separation and are only indicative of the low-boundary concentration regime explored in the current experiments. In particular, turbidity experiments do not provide a measurement of tie-lines, though they are inherently a reflection of the free energy and chemical potential of the solution mixture[73]. Interestingly, phase separation occurs at relatively low concentrations, in the low μM range, which are compatible with physiological concentration of the protein and nucleic acids. Though increasing salt concentration results in an upshift of the phase boundaries, one has to consider that in a cellular environment this effect might be counteracted by cellular crowding.

One peculiar characteristic of our measured phase diagram is the narrow regime of conditions in which we observe phase separation of nonspecific RNA at a fixed concentration of protein. This leads us to hypothesize that the protein may have evolved to maintain tight control of concentrations at which phase separation can (or cannot) occur. Interestingly, when rescaling the turbidity curves as a ratio between protein and RNA, we find all the curve maxima aligning at a similar stoichiometry, approximately 20 nucleotides per protein in the absence of added salt and 30 nucleotides when adding 50 mM NaCl (Fig. S13). These ratios are in line with the charge neutralization criterion proposed by Banerjee et al.[74], since the estimated net charge of the protein at pH 7.4 is +24. Finally, given we observed phase separation with poly(rU), the behavior we are observing is likely driven by relatively nonspecific protein:RNA interactions. In agreement, work from a number of other groups has also established this phenomenon across a range of solution conditions and RNA types[20–27].

Having established phase separation through a number of assays, we wondered what—if any—physiological relevance this may have for the normal biology of SARS-CoV-2.

**A simple polymer model shows symmetry breaking can facilitate multiple metastable single-polymer condensates instead of a single multi-polymer condensate.** Why might phase separation of N protein with RNA be advantageous to SARS-CoV-2? One possible model is that large, micron-sized cytoplasmic condensates of N protein and RNA form through phase separation and facilitate genome packaging. These condensates may act as molecular factories that help concentrate the components for pre-capsid assembly (where we define a pre-capsid here simply as a species that contains a single copy of the genome with multiple copies of the associated N protein), a model that has been proposed in other viruses[75].

However, given that phase separation is unavoidable when high concentrations of multivalent species are combined, we propose

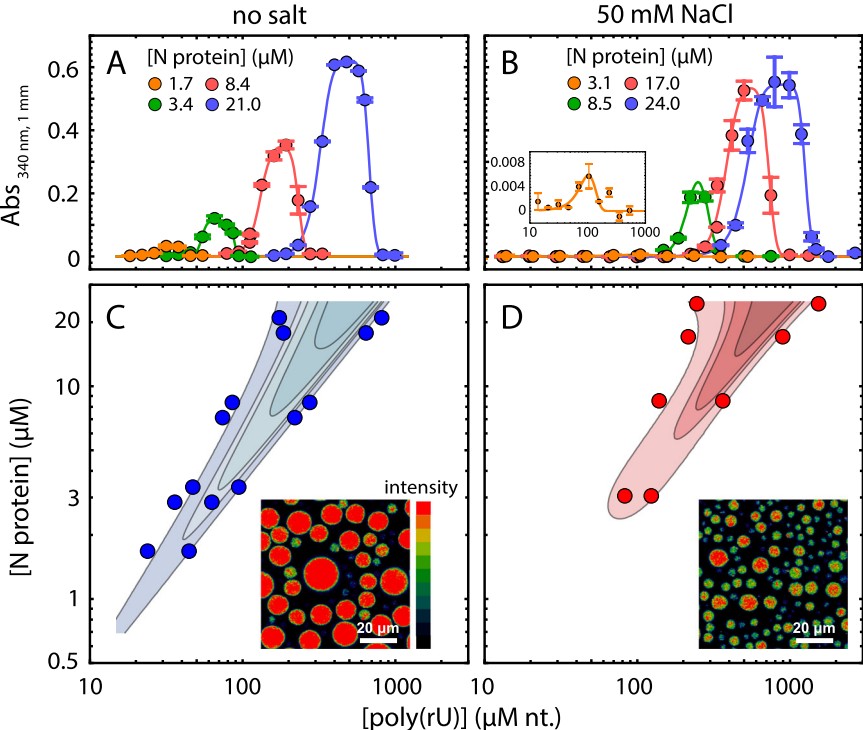

**Fig. 5 Nucleocapsid protein undergoes phase separation with RNA. A**, **B** Appearance of solution turbidity upon mixing was monitored to determine the concentration regime in which N protein and poly(rU) undergo phase separation. Representative turbidity titrations with poly(rU) in 50 mM Tris, pH 7.5 (HCl) at room temperature, in the absence of added salt (**A**) and in the presence of 50 mM NaCl (**B**), at the indicated concentrations of N protein. Points and error bars represent the mean and standard deviation of 2 (absorbance < 0.005) and 4 (absorbance ⩾ 0.005) consecutive measurements from the same sample. Solid lines are simulations of an empirical equation fitted individually to each titration curve (see SI). An inset is provided for the titration at 3.1 µM N protein in 50 mM NaCl to show the small yet detectable change in turbidity on a different scale. **C**, **D** Projection of phase boundaries for poly(rU) and N protein mixtures highlights a re-entrant behavior, as expected for phase separations induced by heterotypic interactions. Turbidity contour lines are computed from a global fit of all titration curves (see SI). Insets: confocal fluorescence images of droplets doped with fluorescently labeled N protein. Total concentrations are 22 µM N protein, 0.5 nM labeled N protein, and 0.54 mM nt. poly(rU). At a higher salt concentration, a lower concentration of protein in the droplet is detected.

that an alternative interpretation of our data is that in this context, phase separation is simply an inevitable epiphenomenon that reflects the inherent multivalency of the N protein for itself and for RNA. This poses questions about the origin of specificity for viral genomic RNA (gRNA), and, of focus in our study, how phase separation might relate to a single-genome packaging through RNA compaction.

Given the expectation of a single genome per virion, we reasoned SARS-CoV-2 might have evolved a mechanism to limit phase separation with gRNA (i.e., to avoid multi-genome condensates), with a preference instead for single-genome packaging (single-genome condensates). This mechanism may exist in competition with the intrinsic phase separation of the N protein with other nonspecific RNAs (nsRNA).

One possible way to limit phase separation between two components (e.g., gRNA/nsRNA and N protein) is to ensure the levels of these components are held at a sufficiently low total concentration such that the phase boundary is never crossed. While possible, such a regulatory mechanism is at the mercy of extrinsic factors that may substantially modulate the saturation concentration[76–78]. Furthermore, not only must phase separation be prevented, but gRNA compaction should also be promoted through the binding of N protein. In this scenario, the affinity between gRNA and N protein plays a central role in determining the required concentration for condensation of the macromole-cule (gRNA) by the ligand (N protein).

Given a system composed of components with defined valencies, phase boundaries are encoded by the strength of

interaction between the interacting domains in the components. Considering a long polymer (e.g., gRNA) with proteins adsorbed onto that polymer as adhesive points (stickers), the physics of associative polymers predicts that the same interactions that cause phase separation will also control the condensation of individual long polymers[69,70,79–82]. With this in mind, we hypothesized that phase separation is reporting on the physical interactions that underlie genome compaction.

To explore this hypothesis, we developed a simple computa-tional model where the interplay between compaction and phase separation could be explored. Our setup consists of two types of species: long multivalent polymers and short multivalent binders (Fig. 6A). All interactions are isotropic, and each bead is inherently multivalent as a result. In the simplest instantiation of this model, favorable polymer:binder and binder:binder interactions are encoded, mimicking the scenario in which a binder (e.g., a protein) can engage in nonspecific polymer (RNA) interaction as well as binder–binder (protein–protein) interac-tion. As expected for simulations of binders with homopolymer polymers we observed phase separation in a concentration-dependent manner (Fig. 6B, E). Phase separation gives rise to a single large spherical cluster with multiple polymers and binders (Fig. 6D, H–L).

Given our homopolymers undergo robust phase separation, we wondered if a break in the symmetry between intra- and intermolecular interactions would be enough to promote single-polymer condensation in the same concentration regime over which we had previously observed phase separation. Symmetry

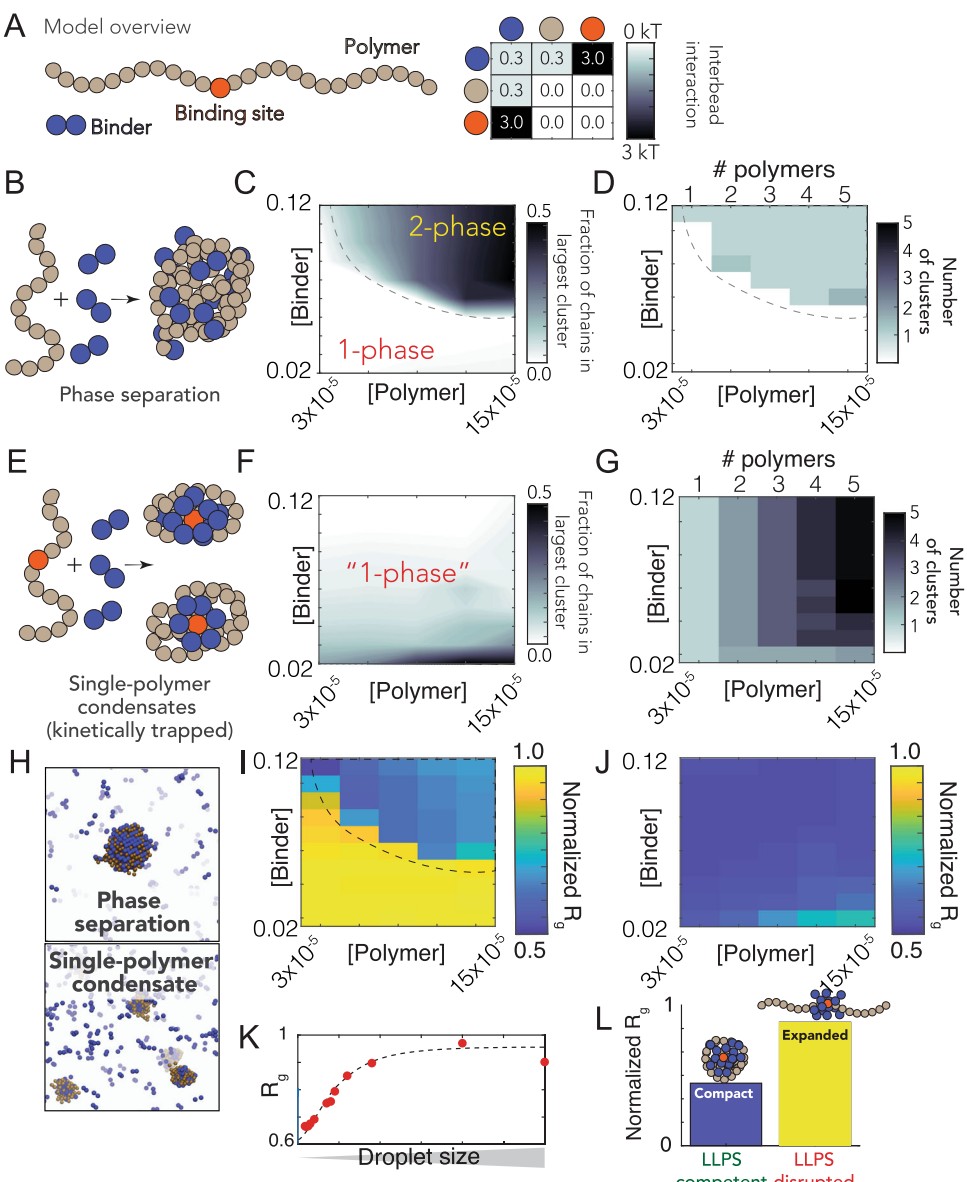

**Fig. 6 A simple polymer suggests symmetry breaking can promote single-polymer condensates over multi-polymer assemblies. A** Summary of our model setup, which involves long polymers (61 beads per molecules) or short binders (2 beads per molecules). Each bead is multivalent and can interact with every adjacent lattice site. The interaction matrix to the right defines the pairwise interaction energies associated with each of the bead types. **B** Concentration-dependent assembly behavior for polymers lacking a high-affinity binding site. Schematic showing polymer architecture (brown) with binder (blue). **C** Phase diagram showing the concentration-dependent phase regime—dashed line represents the binodal (phase boundary) and is provided to guide the eye. **D** Analysis in the same 2D space as panel **C**, assessing the number of droplets at a given concentration. When phase separation occurs, a single droplet appears in almost all cases. **E** Concentration-dependent assembly behavior for polymers with a high-affinity binding site (red bead). **F** No large droplets are formed in any of the systems, although multiple polymer:binder complexes form. **G** The number of clusters observed matches the number of polymers in the system—i.e., each polymer forms an individual cluster. **H** Simulation snapshots from equivalent simulations for polymers with (top) or without (bottom) a single high-affinity binding site. **I** Polymer dimensions in the dense and dilute phase (for the parameters in our model) for polymers with no high-affinity binding site. Note that compaction in the dense phase reflects finite-size effects, as addressed in panel **K**, and is an artifact of the relatively small droplets formed in our systems (relative to the size of the polymer). The droplets act as a bounding cage for the polymer, driving their compaction indirectly. **J** Polymer dimensions across the same concentration space for polymers with a single high-affinity binding site. Across all concentrations, each individual polymer is highly compact. **K** Compaction in the dense phase (panel **I**) is due to small droplets. When droplets are sufficiently large, we observe chain expansion, as expected from standard theoretical descriptions. **L** Simulations performed under conditions in which nonspecific interactions between binder and polymer are reduced (interaction strength = 0 kT). Under these conditions phase separation is suppressed. Equivalent simulations for polymers with a high-affinity site reveal these chains are no longer compact. As such, phase separation offers a readout that—in our model—maps to single-polymer compaction.

breaking in our model is achieved through a single high-affinity-binding site (Fig. 6A). We choose this particular mode of symmetry breaking to mimic the presence of a packaging signal—a region of the genome that is essential for efficient viral packaging—an established feature in many viruses (including coronaviruses) although we emphasize this is a general model, as opposed to trying to directly model gRNA with a packaging signal[83–85].

We performed identical simulations to those in Fig. 6C, D using the same system with polymers that now possess a single high-affinity binding site (Fig. 6E). Under these conditions we did not observe large phase separated droplets (Fig. 6F). Instead, each individual polymer undergoes collapse to form a single-polymer condensate (Fig. 6E). Collapse is driven by the recruitment of binders to the high-affinity site, where they coat the chain, forming a local cluster of binders on the polymer. This cluster is then able to interact with the remaining regions of the polymer through weak nonspecific interactions, the same interactions that drove phase separation in Fig. 6B–D. Symmetry breaking is achieved because the local concentration of binder around the site is high, such that intramolecular interactions are favored over intermolecular interaction. This high local concentration also drives compaction at low binder concentrations. As a result, instead of a single multi-polymer condensate, we observe multiple single-polymers condensates, where the absolute number matches the number of polymers in the system (Fig. 6G).

The high-affinity-binding site polarizes the single-polymer condensate, such that they are organized, recalcitrant to fusion, and kinetically metastable. To illustrate this metastable nature, extended simulations using an approximate kinetic Monte Carlo scheme demonstrated that a high-affinity-binding site dramatically slows assembly of multichain assemblies, but that ultimately these are the thermodynamically optimal configuration (Fig. S18). A convenient physical analogy is that of a micelle, which are non-stoichiometric stable assemblies. Even for micelles that are far from their optimal size, fusion is slow because it requires substantial molecular reorganization and the breaking of stable interactions[86,87].

Finally, we ran simulations under conditions in which binder:polymer interactions were reduced, mimicking the scenario in which nonspecific protein:RNA interactions are inhibited (Fig. 6L). Under these conditions no phase separation occurs for polymers that lack a high-affinity-binding site, while for polymers with a high-affinity-binding site no chain compaction occurs (in contrast to when binder:polymer interactions are present, see Fig. 6J). This result illustrates how phase separation offers a convenient readout for molecular interactions that might otherwise be challenging to measure.

We emphasize that our conclusions from these coarse-grained simulations are subject to the parameters in our model. We present these results to demonstrate an example of how this single-genome packaging could be achieved, offering a class of mechanism that may be in play. This is in contrast to the much stronger statement that this is how it is achieved, a statement that would require much more evidence to make. Recent elegant work by Ranganathan and Shakhnovich[88] identified kinetically arrested microclusters, where slow kinetics result from the saturation of stickers within those clusters. This is completely analogous to our results (albeit with homotypic interactions, rather than heterotypic interactions), giving us confidence that the physical principles uncovered are robust and, we tentatively suggest, quite general. Future simulations are required to systematically explore the details of the relevant parameter space in our system. However, regardless of those parameters, our model does establish that if weak multivalent interactions underlie the formation of large multi-polymer

droplets, those same interactions cannot also drive polymer compaction inside the droplet.

## Discussion

The nucleocapsid (N) protein from SARS-CoV-2 is a multivalent RNA-binding protein critical for viral replication and genome packaging[11,12]. To better understand how the various folded and disordered domains interact with one another, we applied single-molecule spectroscopy and all-atom simulations to perform a detailed biophysical dissection of the protein, uncovering several putative interaction motifs. Furthermore, based on both sequence analysis and our single-molecule experiments, we anticipated that N protein would undergo phase separation with RNA. In agreement with this prediction, and in line with work from the Gladfelter and Yildiz groups working independently from us, we find that N protein robustly undergoes phase separation in vitro with model RNA under a range of different salt conditions. Using simple polymer models, we propose that the same interactions that drive phase separation may also drive genome packaging into a dynamic, single-genome condensate. The formation of single-genome condensates (as opposed to multi-genome droplets) is influenced by the presence of one (or more) symmetry-breaking interaction sites, which we tentatively suggest could reflect packaging signals in viral genomes.

**All three IDRs are highly dynamic**. Our single-molecule experiments and all-atom simulations are in good agreement with one another and reveal that all three IDRs are extended and, depending on solution condition, highly dynamic. Simulations suggest the NTD may interact transiently with the RBD, which offers an explanation for the slightly slowed reconfiguration time measured by nanosecond FCS. The LINK shows rapid rearrangement, demonstrating the RBD and dimerization domain are not interacting. Finally, we see a pronounced interaction between the CTD and the dimerization domain, although these interactions are still highly transient.

Single-molecule experiments and all-atom simulations were performed on monomeric versions of the protein, yet N protein has previously been shown to undergo dimerization and form higher-order oligomers in the absence of RNA[36]. To assess the formation of oligomeric species, we use a combination of NativePAGE, crosslinking, and FCS experiments (see Fig. S14 and SI). These experiments and the comparison between full-length and truncated variants suggest that in the concentration regime used for single-molecule experiments the protein exists as a monomer.

**Simulations identify multiple transient helices**. We identified a number of transient helical motifs that provide structural insight into previously characterized molecular interactions. Transient helices are ubiquitous in viral disordered regions and have been shown to underlie molecular interactions in a range of systems[75,89–91]. While the application of molecular simulations to identify transient helices in disordered regions can suffer from forcefield inaccuracies, it is worth noting that in prior work we have found good agreement between experimental and simulated secondary structure analysis across a range of systems explored in an analogous manner[70,92–94].

Transient helix H2 (in the NTD) and H3 (in the LINK) flank the RBD and organize a set of arginine residues to face the same direction (Figs. 2H and 3F). Both the NTD and LINK have been shown to drive RNA binding, such that we propose these helical arginine-rich motifs (ARMs) may engage in both nonspecific binding and may also contribute to RNA specificity, as has been proposed previously[29,95,96]. The serine–arginine SR region (which

includes H3) has been previously identified as engaging in interaction with a structured acidic helix in Nsp3 in the model coronavirus MHV, consistent with an electrostatic helical interaction[97,98]. Recent NMR data also show excellent agreement with our results, identifying a transient helix that shows 1:1 overlap with H3 [24]. The SR region is necessary for recruitment to replication-transcription centers in MHV, and also undergoes phosphorylation, setting the stage for a complex regulatory system awaiting exploration[99,100].

Transient helix H4 (in the LINK, Fig. 3F) was previously predicted bioinformatically and identified as a conserved feature across different coronaviruses, in agreement with our own secondary structure predictions (Fig. S19)[29]. Furthermore, the equivalent region was identified in SARS coronavirus as a NES, such that we suspect this too is a classical Crm1-binding leucine-rich NES[101]. Jack et al.[20] identified helix H4 as enriched for homotypic cross-links in the context of droplets, supporting a model in which this region promotes protein:protein interactions, an interpretation corroborated by hydrogen–deuterium exchange mass spectrometry on RBD–LINK in the dilute phase[26].

Concerning the CTD, two transient helices are identified, helix H5 and H6. While transient helix H5 is weakly populated, the positive charge associated with this region may make it critical for protein:RNA interaction, a result strongly supported by the observation that deletion of this region ablates protein: RNA phase separation[20]. Transient helix H6 is an amphipathic helix with a highly hydrophobic face (Fig. 4H). Recent hydrogen–deuterium exchange mass spectrometry also identified H6 [41]. Residues in this region have previously been identified as mediating M protein binding in other coronaviruses, such that we propose H6 underlies that interaction[21,102–104]. Recent work has also identified amphipathic transient helices in disordered proteins as interacting directly with membranes, such that an additional (albeit entirely speculative) role could involve direct membrane interaction, as has been observed in other viral phosphoproteins[105,106].

As a final note, while these helices are conserved between SARS, SARS-CoV-2, and in many bat-coronaviruses, they are less well conserved in MHV and MERS, suggesting these regions are malleable over evolution (Fig. S1/3/5).

**The physiological relevance of nucleocapsid protein phase separation in SARS-CoV-2 physiology.** Our work has revealed that SARS-CoV-2 N protein undergoes phase separation with RNA when reconstituted in vitro. The solution environment and types of RNA used in our experiments are very different from the cytoplasm and viral RNA. However, similar results have been obtained in published and unpublished work by several other groups under a variety of conditions, including via in cell experiments[20–27]. Taken together, these results demonstrate that N protein can undergo bona fide phase separation, and that N protein condensates can form in cells. Nevertheless, the complexity introduced by multidimensional linkage effects in vivo could substantially influence the phase behavior and composition of condensates observed in the cell[78,81,107]. Of note, the regime we have identified in which phase separation occurs (Fig. 5) is remarkably relatively narrow, consistent with a model in which single-genome condensates for virion assembly are favored over larger multi-genome droplets.

Does phase separation play a physiological role in SARS-CoV-2 biology? Phase separation has been invoked or suggested in a number of viral contexts to date[108–114]. In SARS-CoV-2, one possible model suggests phase separation may drive recruitment of components to viral replication sites, although how this dovetails with the fact that replication occurs in double-membrane-bound vesicles (DMVs) remains to be explored[24,115]. An alternative (and non-mutually exclusive) model is one in which phase separation catalyzes nucleocapsid polymerization, as has been proposed in elegant work on measles virus[75]. Here, the process of phase separation is decoupled from genome packaging, where gRNA condensation occurs through association with a helical nucleocapsid. If applied to SARS-CoV-2, such a model would suggest that (1) initially N protein and RNA phase separate in the cytosol, (2) some discrete pre-capsid state forms within condensates, and (3) upon maturation, the pre-capsid is released from the condensate and undergoes subsequent virion assembly by interacting with the membrane-bound M, E, and S structural proteins at the ER–Golgi intermediate compartment (ERGIC). While this model is attractive it places a number of constraints on the physical properties of this pre-capsid, not least that the ability to escape the parent condensate dictates that the assembled pre-capsid must interact less strongly with the condensate components than in the unassembled state. This requirement introduces some thermodynamic complexities: how is a pre-capsid state driven to assemble if it is necessarily less stable than the unassembled pre-capsid, and how is incomplete or abortive pre-capsid formation avoided if—as assembly occurs—the pre-capsid becomes progressively less stable?

A phase separation and assembly model raises additional questions, such as the origins of specificity for recruitment of viral proteins and viral RNA, the kinetics of pre-capsid-assembly within a large condensate, and preferential packaging of gRNA over sub-genomic RNA. None of these questions are unanswerable, nor do they invalidate this model, but they should be addressed if the physiological relevance of large cytoplasmic condensates is to be further explored in the context of virion assembly.

Our preferred interpretation is that N protein has evolved to drive genome compaction for packaging (Fig. 7). In this model, a single-genome condensate forms through N protein gRNA interaction, driven by a small number of high-affinity sites. This (meta)-stable single-genome condensate undergoes subsequent maturation, leading to virion assembly. In this model, condensate-associated N proteins are in exchange with a bulk pool of soluble N protein, such that the interactions that drive compaction are heterogeneous and dynamic. Our model provides a physical mechanism in good empirical agreement with data for N protein oligomerization and assembly[116–118]. Furthermore, the resulting condensate is then in effect a multivalent binder for M protein, which interacts with N directly, and may drive membrane curvature and budding in a manner similar to that proposed by Bergeron-Sandoval and Michnick (though with a different directionality of the force) and in line with recent observations from cryo-electron tomography (cryoET)[115,119–121]

An open question pertains to specificity of packaging gRNA while excluding other RNAs. One possibility is for two high-affinity N-protein-binding sites to flank the 5′ and 3′ ends of the genome, whereby only RNA molecules with both sites are competent for compaction. A recent map of N protein binding to gRNA has revealed high-affinity-binding regions at the 5′ and 3′ ends of the gRNA, in good agreement with this qualitative prediction[22]. Alternatively, only gRNA condensates may possess the requisite valency for N protein binding to drive virion assembly through interaction with M protein at the cytoplasmic side of the ERGIC, offering a physical selection mechanism for budding.

Genome compaction through dynamic multivalent interactions would be especially relevant for coronaviruses, which have extremely large single-stranded RNA genomes. This is evolutionarily appealing, in that as the genome grows larger, compaction becomes increasingly efficient, as the effective valence of the

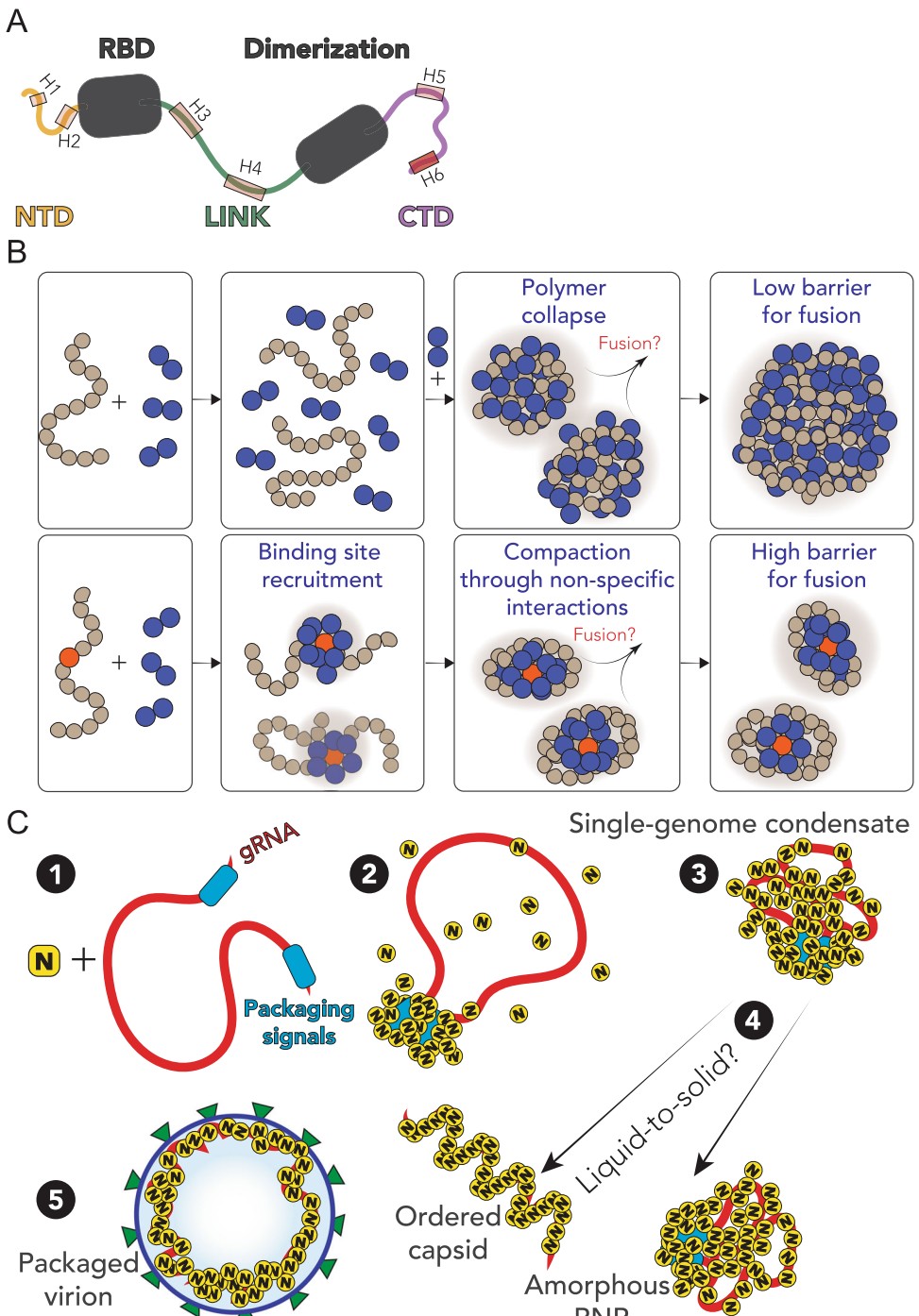

**Fig. 7 Summary and proposed model. A** Summary of results from single-molecule spectroscopy experiments and all-atom simulations. All three predicted IDRs are disordered, highly flexible, and house a number of putative helical-binding regions which overlap with subregions identified previously to drive N protein function. **B** Overview of general symmetry-breaking model. For homopolymers, local collapse leads to single-polymer condensates with a small barrier to fusion, rapidly assembling into large multi-polymer condensates. When one (or a small number of) high-affinity sites are present, local clustering of binders at a lower concentration organize the polymer such that single-polymer condensates are kinetically stable. **C** Proposed model for SARS-CoV-2 genome packaging. (1) Simplified model of SARS-CoV-2 genome with a pair of packaging region at the 5′ and 3′ end of the genome. (2) N protein preferentially binds to packaging signal regions in the genome, leading to a local cluster of N protein at the packaging signal RNA. (3) The high local concentration of N protein drives condensation of distal regions of the genome, forming a stable single-genome condensate. (4) Single-genome condensates may undergo subsequent maturation through a liquid-to-solid (crystallization) transition to form an ordered crystalline capsid, or solidify into an amorphous ribonuclear particle (RNP), or some combination of the two. While in some viruses an ordered capsid clearly forms, we favor a model in which the SARS-CoV-2 capsid is an amorphous RNP. Compact single-genome condensates ultimately interact with E, S, and M proteins at the membrane, whose concerted action leads to envelope formation around the viral RNA and final virion packaging.

genome is increased[69,80]. The ability of multivalent disordered proteins to drive RNA compaction has been observed previously in various contexts[14,122]. Furthermore, genome compaction by RNA-binding protein has been proposed and observed in other viruses[118,123,124], and the SARS coronavirus N protein has previously been shown to act as an RNA chaperone, an expected consequence of compaction to a dynamic single-RNA condensate that accommodates multiple N proteins with a single RNA[14,125]. Furthermore, previous work exploring the ultrastructure of phase separated condensates of G3BP1 and RNA through simulations and cryoET revealed a beads-on-a-string type architecture, mirroring recent results for obtained from cryo-electron tomography of SARS-CoV-2 virions[71,115].

N protein has been shown to interact directly with a number of proteins studied in the context of biological phase separation which may influence assembly in vivo[5,23,70,77,126]. In particular, G3BP1—an essential stress-granule protein that undergoes phase separation—was recently shown to co-localize with overexpressed N protein[24,71,77,127,128]. G3BP1 interaction may be part of the innate immune response, leading to stress-granule formation, or alternatively N protein may attenuates the stress response by sequestering G3BP1, depleting the cytosolic pool, and preventing stress-granule formation, as has been shown for HIV-1 and very recently proposed explicitly for SARS-CoV-2 [112,128].

Our model is also in good empirical agreement with recent observations made for other viruses[129]. Taken together, we speculate that viral packaging may—in general—involve an initial genome compaction through multivalent protein:RNA and protein:protein interactions, followed by a liquid-to-solid transition in cases where well-defined crystalline capsid structures emerge. Liquid-to-solid transitions are well established in the context of neurodegeneration with respect to disease progression[130–132]. Here we suggest nature is leveraging those same principles as an evolved mechanism for monodisperse particle assembly.

Regardless of if phase separated condensates form inside cells, all available evidence suggests phase separation is reporting on a physiologically important interaction that underlies genome compaction (Fig. 6L). With this in mind, from a biotechnology standpoint, phase separation may be a convenient readout for in vitro assays to interrogate protein:RNA interaction. Regardless of which model is correct, N protein:RNA interaction is key for viral replication. As such, phase separation provides a macroscopic reporter on a nanoscopic phenomenon, in line with previous work[70,80,133,134]. In this sense, we propose the therapeutic implications of understanding and modulating phase separation here (and elsewhere in biology) are conveniently decoupled from the physiological relevance of actual, large phase separated liquid droplets, but instead offer a window into the underlying physical interactions that lead to condensate formation[20].

**The physics of single-polymer condensates**. Depending on the molecular details, single-polymer condensates may be kinetically stable (but thermodynamically unstable, as in our model simulations) or thermodynamically stable. Delineation between these two scenarios will depend on the nature, strength, valency, and anisotropy of the interactions. It is worth noting that from the perspective of functional biology, kinetic stability may be essentially indistinguishable from thermodynamic stability, depending on the lifetime of a metastable species.

It is also important to emphasize that at higher concentrations of N protein and/or after a sufficiently long time period we expect robust phase separation with viral RNA, regardless of the presence of a symmetry-breaking site. Symmetry breaking is achieved when the apparent local concentration of N protein (from the perspective of gRNA) is substantially higher than the actual global concentration. As effective local and global concentrations approach one another, the entropic cost of intramolecular interaction is outweighed by the availability of intermolecular partners. On a practical note, if the readout in question is the presence/absence of liquid droplets, a high-affinity site may be observed as a shift in the saturation concentration which, confusingly, could either suppress or enhance phase separation. Further, if single-genome condensates are kinetically stable and driven through electrostatic interactions, we would expect a complex temperature dependence, in which larger droplets are observed at higher temperature (up to some threshold). Recent work is showing a strong temperature dependence of phase separation is consistent with these predictions[22].

Finally, we note no reason to assume single-RNA condensates should be exclusively the purview of viruses. RNAs in eukaryotic cells may also be processed in these types of assemblies, as opposed to in large multi-RNA RNPs. The role of RNA:RNA interactions both here and in other systems is also of particular interest and not an aspect explored in our current work, but we anticipate may play a key role in the relevant biology.

## Methods

**All-atom simulations**. All-atom Monte Carlo simulations were performed with the ABSINTH implicit solvent model (abs_3.2_opls.prm) and CAMPARI simulation engine (V2) (http://campari.sourceforge.net/)[56,135] with the solution ion parameters of Mao et al.[136]. Simulations were performed using movesets and Hamiltonian parameters as reported previously[70,137]. All simulations were performed in sufficiently large box sizes to prevent finite-size effects (where box size varies from system to system). For simulations with IDRs in isolation all degrees of freedom available in CAMPARI are sampled. For simulations with folded domains with IDRs, the backbone dihedral angles in folded domains are not sampled, such that folded domains remain structurally fixed (although sidechains are fully sampled). The IDR has backbone and sidechain degrees of freedom sampled. Simulation sequences used are defined in SI Table S7.

All-atom molecular dynamics simulations were performed using GROMACS (version 5.0.4), using the FAST algorithm in conjunction with the Folding@home platform[57,138,139]. Post-simulation analysis was performed with Enspara[140]. For additional simulation details see the Supplementary Information.

**Coarse-grained polymer simulations**. Coarse-grained Monte Carlo simulations were performed using the PIMMS simulation engine[141]. All simulations were performed in a $70 \times 70 \times 70$ lattice-site box. The results averaged over the final 20% of the simulation to give average values at equivalent states. The polymer species is represented as a 61-residue polymer with either a central high-affinity binding site or not. The binder is a two-bead species. All simulations shown in Fig. 6 were run for $20 \times 10^9$ Monte Carlo steps, with four independent replicas. Bead interaction strengths were defined as shown in Fig. 6A. For additional simulation details see SI.

**Protein expression, purification, and labeling**. SARS-CoV-2 Nucleocapsid protein (NCBI Reference Sequence: YP_009724397.2) including an N term extension containing His$_9$-HRV 3 C protease site was cloned into the *Bam*HI *Eco*RI sites in the MCS of pGEX-6P-1 vector (GE Healthcare). Site-directed mutagenesis was performed on the His$_9$-SARS-CoV-2 Nucleocapsid pGEX vector to create the N protein constructs (SI Table S1) and sequences were verified using Sanger sequencing. All variants were expressed recombinantly in BL21 Codon-plus pRIL cells (Agilent) or Gold BL21(DE3) cells (Agilent) and purified using a FF HisTrap column. The GST-His$_9$-N tag was then cleaved using HRV 3C protease and further purified to remove the cleaved tag. Finally, purified N protein variants were analyzed using SDS-PAGE and verified by electrospray ionization mass spectrometry (LC-MS). Activity of the protein was assessed by testing whether the protein is able to bind and condense nucleic acids (see phase-separation experiments) as well as to form dimers (see oligomerization in SI).

All nucleocapsid variants were labeled with Alexa Fluor 488 maleimide and Alexa Fluor 594 maleimide (Molecular Probes) under denaturing conditions following a two-step sequential labeling procedure (see SI).

**Single-molecule fluorescence spectroscopy**. Single-molecule fluorescence measurements were performed with a Picoquant MT200 instrument (Picoquant, Germany). FRET experiments were performed by exciting the donor dye with a

laser power of 100 μW (measured at the back aperture of the objective). For pulsed interleaved excitation of donor and acceptor, the power used for exciting the acceptor dye was adjusted to match the acceptor emission intensity to that of the donor (between 50 and 70 mW). Single-molecule FRET efficiency histograms were acquired from samples with protein concentrations between 50 and 100 pM and the population with stoichiometry corresponding to 1:1 donor:acceptor labeling was selected. Trigger times for excitation pulses (repetition rate 20 MHz) and photon detection events were stored with 16 ps resolution. For FRET-FCS, samples of double-labeled protein with a concentration of 100 pM were excited by either the diode laser or the supercontinuum laser at the powers indicated above.

All samples were prepared in 50 mM Tris pH 7.32, 143 mM β-mercaptoethanol (for photoprotection), 0.001% Tween 20 (for limiting surface adhesion) and GdmCl at the reported concentrations. All measurements were performed in uncoated polymer coverslip cuvettes (Ibidi, Wisconsin, USA) and custom-made glass cuvette coated with PEG (see SI). Each sample was measured for at least 30 min at room temperature (295 ± 0.5 K).

**Reporting summary**. Further information on research design is available in the Nature Research Reporting Summary linked to this article.

## Data availability

Data supporting the findings in this paper are available from the corresponding authors upon request. All-atom simulation data for Monte Carlo simulations and disorder prediction info are provided at https://github.com/holehouse-lab/supportingdata/tree/master/2021/cubuk_nucleocapsid_2021. Simulations and simulation analysis were performed with open source tools (http://campari.sourceforge.net/, https://camparitraj.readthedocs.io/, http://mdtraj.org/, https://www.gromacs.org/) and Folding@Home data are available for further analysis at https://covid.molssi.org//org-contributions/#folding--home.

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

## Acknowledgements

We thank Amy Gladfelter, Christiane Iserman, Christine Roden, Ahmet Yildiz, Amanda Jack, Luke Ferro, Steve Michnick, Pascale Legault, and Jim Omichinski for sharing data and extensive discussion. We also thank Rohit Pappu for placing our groups in contact with one another. We thank the labs of John Cooper, Carl Frieden, and Silvia Jansen for providing some of the reagents we have used in this work. We thank Ben Schuler and Daniel Nettels for developing, maintaining, and sharing with us the software package used to analyze the single-molecule data. J.C. and J.J.A. are supported by NIGMS R25 IMSD Training Grant GM103757. We are grateful to the citizen-scientists of Folding@home for donating their computing resources. G.R.B. holds an NSF CAREER Award MCB-1552471, NIH R01GM12400701, a Career Award at the Scientific Interface from the Burroughs Wellcome Fund, and a Packard Fellowship for Science and Engineering from The David and Lucile Packard Foundation. A.S. holds NIH grant R01AG062837. A.S.H. is supported by the Longer Life Foundation: an RGA/Washington University Collaboration.

## Author contributions

J.C. designed, expressed, and purified the constructs, performed the single-molecule spectroscopy and oligomerization experiments, analyzed the corresponding data, and wrote the manuscript. J.J.A. performed coarse-grained simulations and wrote the manuscript. J.J.I. performed turbidity experiments and wrote the manuscript. M.D.S. B. designed the constructs for single-molecule spectroscopy experiments, supervised protein expression and purification and oligomerization experiments, and wrote the

manuscript. S.S., M.D.W., M.I.Z. and N.V. set up, curated, analyzed, and managed molecular dynamics simulations on both local resources and the Folding@Home supercomputer. D.G. performed bioinformatic analysis. J.A.W. performed theoretical analysis. G.R.B. acquired funding. K.B.H. wrote the manuscript. A.S. conceived of the study, analyzed data, wrote the manuscript, and acquired funding. A.S.H. conceived of the study, analyzed data, performed and analyzed all-atom Monte Carlo simulations and coarse-grained simulations, wrote the manuscript, and acquired funding. G. R.B., K.B.H., A.S. and A.S.H. jointly supervised the work.

## Competing interests

A.S.H. is a scientific consultant with Dewpoint Therapeutics. This affiliation in no way influenced the content of this study. All other authors declare no competing interests.
