## [Peer Review File · Nature Communications]

REVIEWER COMMENTS

Reviewer #1 (Remarks to the Author):

In this manuscript, the authors use a combination of simulations and single-molecule spectroscopy to study the SARS-CoV-2 N protein. They demonstrate that the protein is dynamic, and can undergo liquid-liquid phase separation with RNA. They then use simulations to propose that these same interactions that drive LLPS could also drive single-genome condensation. Overall, this is interesting, nicely written manuscript. I have just a few suggestions.

1. The authors propose in the discussion that the presence of two high-affinity N-protein binding sites flanking the 5' and 3' ends of the genome might allow for specificity of gRNA packaging, but in their simulations, they only test the effects of a single high-efficiency binding site in the middle. If they are going to propose two high-efficiency binding sites at the ends, it would make sense to run simulations with this configuration.

2. I didn't see an explanation for the specific locations of the fluorophores for the FRET experiments. I think this is especially relevant for their "NTD", which appears to have the second fluorophore within the RNA-binding domain, rather than at the boundary between the IDR and RNA-binding domain.

3. The figures do not all appear in the order that they are discussed in the text. For example, Figure 2E is not mentioned in the text until the discussion, and Figure 2F is not mentioned at all.

4. There are a few statements that could benefit from references:

a. The last sentence of the first results paragraph ("Work on N protein from a range of model coronaviruses has shown...") needs references.

b. In the next paragraph, the authors state that "protein-protein and protein-RNA interaction sites have been mapped to all three disordered regions." I'm not sure that it is accurate that all three contain both protein-protein and protein-RNA interaction sites. Do the authors mean "or"? Either way, this statement needs a reference.

5. In addition to references 20-22, it would be good to add a reference to two other recent manuscripts proposing or examining LLPS by N protein: Carlson et al (PMID: 32637943) and Cascarina et al (PMID: 32562316).

6. Generally in the field the NTD refers to the RNA binding domain, and the CTD refers to the dimerization domain, but the authors use this terminology to refer to the disordered tails. While the authors do define their terms, this may create some reader confusion.

Reviewer #2 (Remarks to the Author):

Cubuk et al. present an interesting study on the dynamics and phase separation behaviors of the nucleocapsid protein of SARS-CoV-2 using a combination of single molecule fluorescence spectroscopy and molecular simulations. Like other nucleocapsid proteins, this protein consists of several folded and disordered domains. The authors employed single molecule FRET and fluorescence correlation spectroscopy (FCS) to show that the N- and C-terminal domains and a linker connecting two folded domains are highly disordered and flexible. MD simulation results agree well with the experimental results in terms of the mean distance between the dye labels. In addition, the authors show that this protein phase separates to form liquid droplets with a model RNA molecule. Simulation results propose that only one genome RNA molecule can be present in a droplet. If this is true, it would have an important implication with the viral genome packaging mechanism.

Overall, the results are presented very well, and given the importance of the understanding of the molecular processes related to SARS-CoV-2, this will be of great interest to readers in various disciplines. However, there are a couple of general issues that should be addressed. First, the results are somewhat diverse and loosely connected. Single molecule experiments characterized disordered domains very well, but are virtually unrelated to the phase separation, which is mostly explained by simulations and models. Second, the explanation of the phase separation part and modeling is lengthy and describes too many different things and speculations briefly that cannot be really proved in this study. The authors need to consider making this part more concise. In addition to these, there are more specific questions as listed below, which should also be properly addressed before publication.

1. It looks like there is no cysteine residue in the sequence other than mutations for dye-labeling. Is this correct?

2. In Fig. S7, what is the polymer model (Gaussian or SAW) used for the theoretical lines? In addition, the last sentence in the caption is not clear. How is the overlap between the donor and acceptor distribution defined and why does larger overlap (probably NTD?) indicate possible static distribution?

3. In Fig. S11, the amplitude of the acceptor-donor cross-correlation is negligible for CTD, whereas correlations appear in donor-donor and acceptor-acceptor auto-correlation. This discrepancy was attributed to possible donor fluorescence quenching. Then, which residue in the sequence is responsible for quenching? In addition, fluorophores were not attached site-specifically, and therefore there should be two species (quenched and unquenched or donor quenched and acceptor quenched if acceptor can be quenched as well) unless the quenching efficiency of both sites is the same. Finally, quenching will reduce the fluorescence lifetime. Is this observed? This may also affect the analysis in Fig. S7. Finally, It would be also worth presenting the amplitude of the correlation with the decay time for comparison.

4. In the second paragraph on page 5, "This behavior can be understood assuming that the plateau between 1 M and 2 M GdmCl represents the average of transfer efficiencies between two populations in equilibrium"

It would be better to explicitly say the folded and unfolded populations of RBD instead of "two populations" because it is not immediately clear what the two populations indicate at this point. It took some time to figure this out by reading SI.

5. In the third paragraph on page 5, "Secondly, the RBD contributes significantly to the conformations of the measured NTD construct, mainly by reducing the accessible space of the disordered tail and favoring expanded configurations, as shown by the shift in transfer efficiency when the RBD is unfolded."

The FRET efficiency change may not be explained by accessible space because the situation is similar for the other two disordered domains. Instead, in Fig. 1A, position 68 is a part of the RBD rather than the end of NTD. Could this be a simple reason that the FRET efficiency of NTD is more affected by folding/unfolding of RBD?

6. On page 11, "Given the high interaction valency and the presence of molecular features similar to other proteins we had previously studied"

It would help readers understand if "molecular features" are described a little more specifically.

7. Most of the experiments were performed at 50 mM Tris. This is quite low ionic strength compared to physiological conditions. How do the authors predict the behaviors observed in this work (disordered domains and phase separation) would change at much higher ionic strength (> 100 mM)?

8. Is there any way to experimentally detect single-genome condensate?

9. Have phase separation and viral genome compaction mediated by specific binding sites been

observed or is this a new idea? To make this model work, how different should the specific and non-specific interactions in terms of the ratio of the affinity?

10. What is the time scale of the transient helix formation? Is it possible to measure this using the nanosecond FCS experiment?

11. Linker domain does not interact with both folded domains, but helicity is affected greatly by including folded domains in the simulation. On the other hand, NTD may interact with RBD, but the helicity is not affected at all by including RBD in the simulation. Is there any plausible explanation?

Reviewer #3 (Remarks to the Author):

In this manuscript, Cubuk and co-workers report on a large scale, interdisciplinary study to elucidate the structure and dynamics of the SARS-Cov-2 nucleocapsid protein. This work represents a major advance in our understanding of a protein that is crucial in genome packaging of the SARS-Cov-2 virus, and is therefore a potential target of drugs to treat COVID-19. Overall, the authors have carried out a careful study using complementary biophysical approaches resulting in a unified model for the structure/function of the N protein. The manuscript text is very well-written and the figures are exceptionally clear. I strongly feel that this work should be published with some minor revisions for clarity and completeness, and that it will be a meaningful contribution to the rapidly evolving literature on SARS-Cov-2 proteins.

I have the following suggestions for minor revisions to the manuscript:

(1) page 2, paragraph 2 — What the authors mean by “symmetry breaking” could be more clearly described at this point in the paper.

(2) page 2, paragraph 2 — Including a brief description of refs 20-22 — are the other recent studies of N protein consistent with the present work in any other respects beyond phase separation?

(3) page 2, paragraph 3 — The last sentence of this paragraph would require references.

(4) page 3, Figure 1, panel A — The method used for disorder prediction should be indicated. It would also be interesting and potentially useful to include a prediction of alpha-helicity (and see if this is consistent with the simulation results).

(5) page 4, paragraph 4 — What does “native conditions” refer to?

(6) page 5, paragraph 4 & page 6, Fig 2D— The authors refer to the agreement between the smFRET inferred distance and the $p(r)$ histogram as “excellent” in the text, and as “good” in the figure caption. In my view, “excellent” is too strong an adjective to use in this case, given that there is no direct comparison being carried out, and the fluorescent dyes are ignored.

(7) page 19, paragraph 1 — The conclusion that there are helices present in the disordered regions relies on the results of all-atom simulations. It would be useful to acknowledge the challenge in accurately predicting ensembles of disordered regions using simulations, and the lack of consensus in the field currently on the optimal approach to obtain these ensembles. This may be similar to the discussion on page 17, paragraph 3 where the authors state “We emphasize that our conclusions from simulations are subject to the parameters in our model” — the conclusion about helix population may be highly dependent on force field.

(8) page 19, paragraph 3 — How similar is the sequence of SARS-Cov-2 N protein in the region of

helix 4? — a multiple sequence alignment would provide supporting evidence here.

(9) supporting info, page 2, paragraph 3 — The crystal structure 6VYO is not monomeric, whereas 6YI3, the NMR ensemble, is. The authors should clarify what they mean by “equivalent” here. Which chain from 6VYO is used? Why is this structure used instead of 6YI3?

(10) supporting info, page 4, paragraph 2 — Is the same force field used for both the NTD and RBD simulations? More details are needed here to ensure reproducibility. Providing input files for the simulations, either as supporting info or in an online repository would be useful for reproducibility as well.

We would like to thank the reviewers for their careful reading of the manuscript and insightful comments. Following their suggestion, we have shortened part of the polymer model description in the second part of the paper and improved the description of single-molecule experiments and results. We think the changes we have introduced in the manuscript have increased readability, cohesion, and interpretation of experiments and simulations that will be of interest to a broad set of the scientific community.

In reviewing single-molecule FRET data, we discovered a coding error in the data analysis that was preventing the code from applying the correction factors that account for detection efficiency and quantum yield of the dyes. We apologize for this inconvenience and we have now updated all the corresponding graphs and values in the manuscript. This correction leads to some alterations in the stoichiometry ratio selection and in the corresponding thresholds, resulting in overall better statistics of the same data sets. For what concerns the actual distribution, only small alterations in the transfer efficiency data points are introduced (within a maximum of 0.04 toward higher values). Since the transfer efficiency change is small and largely systematic, little variations are detected for all the connected parameters and none of our conclusions is affected by this error. All data have been corrected accordingly and all the values across the text and tables have been updated. As possible, this correction has improved the agreement between experimental data and simulations. We apologize for the inconvenience.

Inspired by one of the questions of the reviewers regarding the role of the folded RBD in the protein, we have now included a new single-molecule dataset that addresses this point by directly quantifying the folding of the RBD domain. We have also added new single-molecule data for truncated variants of the proteins that allow us to deconvolve the impact of intra-domain interactions on the disordered regions. Specifically, we investigated:

1. The same NTD labeling positions in a protein fragment that contains only the NTD and RBD domains
2. The LINK labeling positions in absence of the Dimer and CTD
3. The CTD labeling positions in a fragment of the CTD (all constructs are reported in **Table S1**).

By investigating these new constructs we have uncovered additional insights:

1. We observe negligible conformational changes in the NTD region regardless of if it is part of the full-length construct (NTD-FL) or in the truncated version of the protein (NTD-RBD), supporting our original conclusions that there are at most weak long-range intramolecular interactions.
2. We identified significant electrostatic interactions in the LINK region when the DIMER and CTD domains are missing, suggesting either a self-interaction or an interaction with the RBD. These observations led us to previous measurements to ensure precise control

of the ion concentration in the final solution after diluting the protein from high denaturant. In doing so, we discovered an interesting second population for the LINK within the full-length protein (LINK FL) that suggests interactions of the LINK with one of the two surrounding domains, possibly the RBD (though interaction with DIMER cannot be excluded). A description of this effect is now accounted in the main text corresponding section and in the SI (see *Salt dependence* section).

3. We compared the CTD-FL with a truncated version of the CTD, obtaining similar properties and conformational changes, though the CTD-FL exhibits a large distribution of transfer efficiencies suggesting that the structural distance arises from interaction with the surrounding folded domains. Finally, we have also extended the ns-FCS measurements to higher concentrations of GdmCl.

Finally, we have also included new simulation data for the LINK and CTD (CTD now follows sampling protocol as was used for NTD), including extensive all-atom molecular dynamics to obtain a diverse ensemble of starting structures for the CTD. These results are also in better agreement with our smFRET data.

The totality of these data enriches our understanding of the nucleocapsid protein and provides what we feel now is a high resolution and high-confidence assessment of the proteins' conformational behavior and dynamics.

In addition, we have performed extensive additional coarse-grained simulations to more explicitly motivate the metastable nature of the single-polymer condensates observed in our simulations. While we remain somewhat agnostic as to whether or not these proposed single polymer condensates are kinetically or thermodynamically stable for real systems, it is important to unambiguously make clear the physical behavior explored in the context of our model system.

Please find our specific point-by-point comments to the reviewers below.

REVIEWER COMMENTS

Reviewer #1 (Remarks to the Author):

In this manuscript, the authors use a combination of simulations and single-molecule spectroscopy to study the SARS-CoV-2 N protein. They demonstrate that the protein is dynamic, and can undergo liquid-liquid phase separation with RNA. They then use simulations to propose that this same interactions that drive LLPS could also drive single-genome condensation. Overall, this is interesting, nicely written manuscript. I have just a few suggestions.

We appreciate the reviewer's positive assessment of our work.

1. The authors propose in the discussion that the presence of two high-affinity N-protein binding sites flanking the 5' and 3' ends of the genome might allow for specificity of gRNA packaging, but in their simulations, they only test the effects of a single high-efficiency binding site in the middle. If they are going to propose two high-efficiency binding sites at the ends, it would make sense to run simulations with this configuration.

This is a point that we considered and discussed at length. The goal of the single-genome condensate theory/simulation work is to propose a 'class' of multivalent assemblies that can form. Our work here is essentially providing a toy system in which intermolecular symmetry is broken. This relates directly to the N protein assemblies inasmuch as we believe there too high-affinity interactions with certain RNA motifs can (within certain concentration regimes) favor single-genome condensates. However, we are deliberately avoiding our simulations from trying to reproduce the actual physical behavior for RNA:N protein because we fundamentally don't yet know enough about how the system works. Specifically, we speculate that there are in fact multiple "packaging signals" distributed across the genome that may vary in affinity, sequence, and structure. As such, explicitly building a model that attempts to recapitulate the genomic architecture of the SARS-CoV-2 genome feels like an attempt to ascribe a level of accuracy and detail to our model that we are not confident in.

The proposed binding sites that emerge from Amy Gladfelters work are certainly consistent with what we would expect, but are one of *many* possible molecular topologies that 'could work'. With this in mind, our simulations are designed to be the simplest type of system where single-polymer condensates can form without adding additional parameters (i.e. varying number/strength/patterning of binding sites, all of which influence the underlying assembly behavior as we have explored in unpublished work). We, therefore, don't wish to try and recapitulate the double-binding site model that may emerge from the Gladfelter work in essence because it would be focussing on one specific feature while conveniently ignoring many others that may also matter a great deal. For context, we have run these simulations and they can form single-molecule condensates, but, in essence, no other outcome is possible within the regime we're exploring so this is a quite misleading result. We have updated the text to make this point clear.

2. I didn't see an explanation for the specific locations of the fluorophores for the FRET experiments. I think this is especially relevant for their "NTD", which appears to have the second fluorophore within the RNA-binding domain, rather than at the boundary between the IDR and RNA-binding domain.

We thank the reviewer for helping us improve the clarity of the text. We have now included a paragraph in the Supplementary Information titled "Choice of labeling positions" that describes the criteria that have guided our choices.

3. The figures do not all appear in the order that they are discussed in the text. For example, Figure 2E is not mentioned in the text until the discussion, and Figure 2F is not mentioned at all.

We thank the Reviewer for pointing out this inconsistency in the presentation of our data and we have now updated the text so that all the figures are mentioned in the right order, and all panels are mentioned.

4. There are a few statements that could benefit from references:

a. The last sentence of the first results paragraph (“Work on N protein from a range of model coronaviruses has shown...”) needs references.

We thank the reviewer for pointing out this. We have now added references to support our statement.

b. In the next paragraph, the authors state that “protein-protein and protein-RNA interaction sites have been mapped to all three disordered regions.” I’m not sure that it is accurate that all three contain both protein-protein and protein-RNA interaction sites. Do the authors mean “or”? Either way, this statement needs a reference.

We thank the review and have referenced and refined the associated language.

5. In addition to references 20-22, it would be good to add a reference to two other recent manuscripts proposing or examining LLPS by N protein: Carlson et al (PMID: 32637943) and Cascarina et al (PMID: 32562316).

Indeed, even since these reviews were received more have emerged, and we believe at the time of submission we have included all relevant references including updating any previously pre-printed papers to their more recently accepted status.

6. Generally in the field the NTD refers to the RNA binding domain, and the CTD refers to the dimerization domain, but the authors use this terminology to refer to the disordered tails. While the authors do define their terms, this may create some reader confusion.

We appreciate the reviewer’s comments here. We have chosen to maintain our current naming convention in part because we feel it makes delineation of the distinct domains more straight forward and follows standard nomenclature in other proteins in which NTD and CTD are frequently used in the context of N-terminal or C-terminal IDRs. We discussed this point at length and considered alternatives, but feel this is the right choice.

Reviewer #2 (Remarks to the Author):

Cubuk et al. present an interesting study on the dynamics and phase separation behaviors of the nucleocapsid protein of SARS-CoV-2 using a combination of single molecule fluorescence spectroscopy and molecular simulations. Like other nucleocapsid proteins, this protein consists of several folded and disordered domains. The authors employed single molecule FRET and fluorescence correlation spectroscopy (FCS) to show that the N- and C-terminal domains and a linker connecting two folded domains are highly disordered and flexible. MD simulation results agree well with the experimental results in terms of the mean distance between the dye labels. In addition, the authors show that this protein phase separates to form liquid droplets with a

model RNA molecule. Simulation results propose that only one genome RNA molecule can be present in a droplet. If this is true, it would have an important implication with the viral genome packaging mechanism.

Overall, the results are presented very well, and given the importance of the understanding of the molecular processes related to SARS-CoV-2, this will be of great interest to readers in various disciplines. However, there are a couple of general issues that should be addressed.

We appreciate the reviewer's positive assessment of our work!

First, the results are somewhat diverse and loosely connected. Single molecule experiments characterized disordered domains very well, but are virtually unrelated to the phase separation, which is mostly explained by simulations and models.

We thank the reviewer for this important point. We have added an additional paragraph to better connect the two sections more clearly, and in the discussion made further references here. In the interest of avoiding further lengthening the already verbose manuscript, we have not significantly extended this text but if this remains a major concern we can do so.

Second, the explanation of the phase separation part and modeling is lengthy and describes too many different things and speculations briefly that cannot be really proved in this study. The authors need to consider making this part more concise.

We agree, we have condensed this section and moved a substantial set of text from discussion to the SI. We will also say that a number of colleagues have reached to share their appreciation for this section as illustrative of broadly relevant ideas, such that while we recognize the section is modeling-based, we believe it has introduced some new ideas that resonate with at least a subset of our readers.

In addition to these, there are more specific questions as listed below, which should also be properly addressed before publication.

1. It looks like there is no cysteine residue in the sequence other than mutations for dye-labeling. Is this correct?

Yes, there is no cysteine residue in the wild-type sequence and the only cysteine residues are inserted in the specific labeling positions.

2. In Fig. S7, what is the polymer model (Gaussian or SAW) used for the theoretical lines? In addition, the last sentence in the caption is not clear. How is the overlap between the donor and acceptor distribution defined and why does larger overlap (probably NTD?) indicate possible static distribution?

We thank the reviewer for helping us improve the clarity of the text. The polymer model used in Fig. S7 is a Gaussian chain model. However, a SAW is indistinguishable in the result predicted for a dynamic line using a Gaussian chain. The discrepancy between flexible polymer models is visible only when the distance distribution approaches the rod-like limit, e.g. in the case of a

wormlike chain with a long persistence length (see Soranno et al, PNAS, 2012; Holmstrom et al, Method. Enz. 2018). The last sentence points to the fact that a fraction of the overall population at a given transfer efficiency may sit closer to the straight black line that represents static conformations. The fact that both acceptor and donor distributions have shifted in a similar direction is compatible with the contact formation between IDR and folded domains observed in the simulations.

In Fig. S11, the amplitude of the acceptor-donor cross-correlation is negligible for CTD, whereas correlations appear in donor-donor and acceptor-acceptor auto-correlation. This discrepancy was attributed to possible donor fluorescence quenching. Then, which residue in the sequence is responsible for quenching? In addition, fluorophores were not attached site-specifically, and therefore there should be two species (quenched and unquenched or donor quenched and acceptor quenched if acceptor can be quenched as well) unless the quenching efficiency of both sites is the same. Finally, quenching will reduce the fluorescence lifetime. Is this observed? This may also affect the analysis in Fig. S7. Finally, It would be also worth presenting the amplitude of the correlation with the decay time for comparison.

We attribute the amplitude in donor-donor and acceptor-acceptor to possible donor and/or acceptor quenching. Indeed a correlation decay is observed in both fluorophores. The reviewer is correct that most likely different quenching efficiencies are at play depending on the labeling positions and we think this is one of the factors that further contributes to the large distribution in transfer efficiencies of the CTD FL. When compared to the CTD fragment one can clearly see that the broadening of the peak interests both sides of the distribution, which is compatible with populations where either the donor or the acceptor are (at least) partially quenched.

Regarding which residues are responsible for quenching, we want to note that the major quencher for Alexa 488 and 594 is tryptophan and no tryptophan residues are found in the CTD sequence. Instead, the most likely source of quenching is the two tryptophan residues in the DIMER domain, in positions 301 and 330. Tyrosine residues are another possible quencher, though have been shown to be weaker than tryptophans. Four tyrosines decorate the DIMER domain in positions 268, 297, 333, and 360. We have started investigating these effects from the closest possible quencher (Y360) but we have detected no change in ns-FCS. We plan to test the effect of other mutations on the conformations of the CTD FL in future experiments.

For what concerns the fluorescence lifetimes (now included in Fig. S7) we have not found significant quenching effects. However, this is not completely surprising and does not conflict with the proposed interpretation. Previous experiments (see Zosel et al. J. Chem Phys 2017) clearly show that, even in a disordered region, the contribution of quenching from an aromatic residue such as W can be significant for static quenching, but barely visible in the lifetime of the molecule. This reflects the fact that static quenching relies on the formation of a transient dark complex between the dye and the quencher, which is often in the order of tens or even hundreds of nanoseconds. Whereas the amplitude depends on the fraction of formed contacts, the contribution of static quenching is made more apparent by the long-lived nature of these

“dark” complexes. The correlation curve captures these dark complexes as a delay in the probability of observing a second donor (or acceptor) photon after having observed a first one. Importantly, these static quenching dark complexes do not contribute to the fluorescence lifetime, since they are dark and not emitting for a longer time than the laser pulse frequency. This is the reason lifetime is sensitive only to dynamic quenching from quencher-dye collisions that do not result in the formation of a static complex. Furthermore, the measured lifetime reflects only the average lifetime weighted for the unquenched and quenched configurations. Depending on the mechanism leading to quenching, if this originates from contact formation, the fraction of contacts is usually a small percentage compared to the overall distribution sampled by the disordered region.

Amplitudes of the CTD correlation are now included in the same figure (Fig. S12, previously named Fig. S11).

4. In the second paragraph on page 5, “This behavior can be understood assuming that the plateau between 1 M and 2 M GdmCl represents the average of transfer efficiencies between two populations in equilibrium ...”

It would be better to explicitly say the folded and unfolded populations of RBD instead of “two populations” because it is not immediately clear what the two populations indicate at this point. It took some time to figure this out by reading SI.

We thank the reviewer and we have corrected the main text accordingly. The paragraph now includes the sentence: “These two populations reflect the contributions of the folded and unfolded configurations of RBD to the labeled NTD segment”.

5. In the third paragraph on page 5, “Secondly, the RBD contributes significantly to the conformations of the measured NTD construct, mainly by reducing the accessible space of the disordered tail and favoring expanded configurations, as shown by the shift in transfer efficiency when the RBD is unfolded.”

The FRET efficiency change may not be explained by accessible space because the situation is similar for the other two disordered domains. Instead, in Fig. 1A, position 68 is a part of the RBD rather than the end of NTD. Could this be a simple reason that the FRET efficiency of NTD is more affected by folding/unfolding of RBD?

We agree with the reviewer that folding of the portion of the segment between position 50 and position 68 can explain the observation and we include this in our explanation of the experimental data:

“We interpret these two populations as the contribution of the folding and unfolding fraction of the RBD domain on the distances probed by the NTD-FL construct, which includes a labeling position within the folded RBD” (page 5)

However, we would like to underline that each domain impacts differently the disordered regions as shown for the LINK and CTD when comparing full and truncated versions. In this respect, the comparison between full length and truncated versions provide us insights into the nature of these interactions.

6. On page 11, “Given the high interaction valency and the presence of molecular features similar to other proteins we had previously studied”

It would help readers understand if “molecular features” are described a little more specifically.

Thanks - we have updated the text here.

7. Most of the experiments were performed at 50 mM Tris. This is quite low ionic strength compared to physiological conditions. How do the authors predict the behaviors observed in this work (disordered domains and phase separation) would change at much higher ionic strength (> 100 mM)?

In Fig. S11 we have explored the effects of KCl (mimicking one of the most abundant ions in cells) up to 500 mM, including 50 mM (for a total ionic strength of 100 mM). Most of the measured points as a function of GdmCl include more measurements at low GdmCl concentrations (below 0.5M), which also act prevalently as a salt (see a comparison with urea in Fig. S10). Overall, the effect of salt seems to promote flexibility and expansion of the LINK region. Future experiments will address possible long-range implications as well as the contribution of ionic strength to oligomerization.

Similarly, for phase separation experiments we have reported examples at 50 mM Tris and 50 mM Tris + 50 mM NaCl (total ionic strength of 100 mM), which suggests that increasing salt concentration shifts the boundaries of phase separation toward higher concentrations of nucleic acid and protein. Therefore, we think that our experiments already prove that similar observations hold even at higher salt concentrations, both in terms of protein conformations and propensity for phase separation. Future works will aim to address this interplay but requires first addressing their influence in the binding of nucleic acid, which we think is beyond the scope of this work.

8. Is there any way to experimentally detect single-genome condensate?

Yes, this is a fantastic question. This is something we are approaching using a combination of other techniques, including force spectroscopy, fluorescence imaging, and negative stain EM. However, we think this goes beyond the scope of the present work.

9. Have phase separation and viral genome compaction mediated by specific binding sites been observed or is this a new idea? To make this model work, how different should the specific and non-specific interactions in terms of the ratio of the affinity?

This is an outstanding question. As far as we know this is a new idea - we are actively exploring this exact idea through both ensemble and single-molecule experiments and theory and simulations.

10. What is the time scale of the transient helix formation? Is it possible to measure this using the nanosecond FCS experiment?

Due to the nature of our molecular simulations, we do not obtain information on chain dynamics (i.e. our all-atom Monte Carlo simulation lacks a time dimension) so cannot easily answer this question from the computation. Technically speaking, this question can be addressed using nanosecond FCS. However, because of the short span of the helix, this would require a completely new design of the experiment centered on Photon-Electron-Transfer PET ns-FCS instead of FRET-FCS. This would require adding cysteines across the CTD in different positions and test the change in PET. We think this is a fantastic suggestion but it is beyond the scope of this work.

11. Linker domain does not interact with both folded domains, but helicity is affected greatly by including folded domains in the simulation. On the other hand, NTD may interact with RBD, but the helicity is not affected at all by including RBD in the simulation. Is there any plausible explanation?

As we mentioned in the introduction, we have now carefully revised the low ionic strength dependence of the LINK construct and identified the appearance of two populations in equilibrium at low salt in presence of the two domains, whereas a single population strongly compacted by electrostatic is present in a truncated version of the protein. This suggests that helicity, as well as transient interaction of the LINK with itself or with the RBD, are strongly affected by the addition of the DIMER domain. This is consistent with also a weak dependence on denaturant, such as the folded domains surrounding the linker are “neutralizing” the contribution of denaturant because are already favoring extended conformations. In a more general picture, we can consider two effects at play: one due to electrostatics (or more in general interactions) and one due to excluded volume effects. Interactions can either attract or repel the disordered region, whereas excluded volume effects are purely repulsive and will limit the possible conformations. The extent of each of these contributions is dictated by the specific nature of the IDR sequence. One particular case that differentiates the LINK from the other constructs is the presence of not one, but two surrounding domains. In this case, the chain configurations are strongly influenced by the properties of the two domains, whereas for NTD and CTD the free N- or C- terminal end of the chain can compensate for the effects.

Reviewer #3 (Remarks to the Author):

In this manuscript, Cubuk and co-workers report on a large scale, interdisciplinary study to elucidate the structure and dynamics of the SARS-Cov-2 nucleocapsid protein. This work represents a major advance in our understanding of a protein that is crucial in genome packaging of the SARS-Cov-2 virus, and is therefore a potential target of drugs to treat COVID-19. Overall, the authors have carried out a careful study using complementary biophysical approaches resulting in a unified model for the structure/function of the N protein. The manuscript text is very well-written and the figures are exceptionally clear. I strongly feel that this work should be published with some minor revisions for clarity and completeness, and

that it will be a meaningful contribution to the rapidly evolving literature on SARS-Cov-2 proteins.

I have the following suggestions for minor revisions to the manuscript:

(1) page 2, paragraph 2 — What the authors mean by “symmetry breaking” could be more clearly described at this point in the paper.

We have attempted to better clarify our thoughts here.

(2) page 2, paragraph 2 — Including a brief description of refs 20-22 — are the other recent studies of N protein consistent with the present work in any other respects beyond phase separation?

Since our initial submission, a number of additional studies have been pre-printed, all of which are in good agreement with our work. We have included additional discussion where relevant on several of these studies in the discussion (including the original refs. 20-22).

(3) page 2, paragraph 3 — The last sentence of this paragraph would require references.

We have updated this reference.

(4) page 3, Figure 1, panel A — The method used for disorder prediction should be indicated. It would also be interesting and potentially useful to include a prediction of alpha-helicity (and see if this is consistent with the simulation results).

The method on disorder prediction is now included in the figure caption, as well as in the updated supplementary information. We have also performed the secondary structure prediction which is now Fig. S18, finding our simulations find experimentally-validated helices that were not predicted using standard secondary-structure prediction tools.

(5) page 4, paragraph 4 — What does “native conditions” refer to?

We have now substituted “native” with “buffer” or “aqueous buffer” conditions.

(6) page 5, paragraph 4 & page 6, Fig 2D— The authors refer to the agreement between the smFRET inferred distance and the $p(r)$ histogram as “excellent” in the text, and as “good” in the figure caption. In my view, “excellent” is too strong an adjective to use in this case, given that there is no direct comparison being carried out, and the fluorescent dyes are ignored.

Very true - good point, have fixed

(7) page 19, paragraph 1 — The conclusion that there are helices present in the disordered regions relies on the results of all-atom simulations. It would be useful to acknowledge the challenge in accurately predicting ensembles of disordered regions using simulations, and the lack of consensus in the field currently on the optimal approach to obtain these ensembles. This may be similar to the discussion on page 17, paragraph 3 where the authors state “We emphasize that our conclusions from simulations are subject to the parameters in our model” — the conclusion about helix population may be highly dependent on force field.

This is an important point and we have added an associated caveat in the discussion.

(8) page 19, paragraph 3 — How similar is the sequence of SARS-Cov-2 N protein in the region of helix 4? — a multiple sequence alignment would provide supporting evidence here.

The associated multiple sequence alignment is provided in Fig. S3, but this is a good point and actually, something we have added into the discussion; while the helices are conserved between SARS and SARS-CoV-2 they are not as strongly conserved in MERS

(9) supporting info, page 2, paragraph 3 — The crystal structure 6VYO is not monomeric, whereas 6YI3, the NMR ensemble, is. The authors should clarify what they mean by “equivalent” here. Which chain from 6VYO is used? Why is this structure used instead of 6YI3?

We have provided clarification here, and to directly answer the reviewer’s question, when the project began (March 13th 2020) 6VYO was the only available structure of the SARS-CoV-2 RBD.

(10) supporting info, page 4, paragraph 2 — Is the same force field used for both the NTD and RBD simulations? More details are needed here to ensure reproducibility. Providing input files for the simulations, either as supporting info or in an online repository would be useful for reproducibility as well.

We have clarified in the SI tex, but all Monte Carlo simulations were performed using the same forcefield and same simulations parameters (standard ABSINTH model available with CAMPARI). The complete keyfile for running simulations is provided here https://github.com/holehouse-lab/supportingdata/tree/master/2020/cubuk_nucleocapsid_2020.

REVIEWERS' COMMENTS

Reviewer #1 (Remarks to the Author):

This manuscript examines LLPS of the SARS-CoV-2 N protein with RNA. While multiple other papers have examined N protein LLPS, this manuscript uses simulations to address a novel hypothesis: that LLPS provides a mechanism for genome condensation.

Overall, the experiments/simulations are well done, the manuscript addresses an important topic that should be of broad interest, and the authors appropriately addressed the reviewer concerns from the first submission.

Reviewer #2 (Remarks to the Author):

The authors have done a great job in responding to my comments. I don't have any further questions and I think the manuscript is ready for publication.

Reviewer #3 (Remarks to the Author):

The authors have fully addressed the comments that I provided in the first review of the manuscript. Furthermore, they have significantly improved the clarity of the manuscript and carefully responded to all revisions suggested by the other two reviewers. The results of their simulations and experiments will be of interest to a broad audience spanning multiple scientific disciplines. I would recommend that the manuscript is accepted for publication in its present form.